# RoboOmni: Proactive Robot Manipulation in Omni-modal Context

**Siyin Wang**[1,2]  **Jinlan Fu**[1†] **Feihong Liu**[1] **Xinzhe He**[1] **Huangxuan Wu**[1]
**Junhao Shi**[1,2] **Kexin Huang**[1] **Zhaoye Fei**[1]
**Jingjing Gong**[2] **Zuxuan Wu**[1,2] **Yu-Gang Jiang**[1] **See-Kiong Ng**[3] **Tat-Seng Chua**[3] **Xipeng Qiu**[1,2†]

[1]Fudan University [2]Shanghai Innovation Institute [3]National University of Singapore

siyinwang20@fudan.edu.cn, jinlanjonna@gmail.com

## ABSTRACT

Recent advances in Multimodal Large Language Models (MLLMs) have driven rapid progress in Vision–Language–Action (VLA) models for robotic manipulation. Although effective in many scenarios, current approaches largely rely on explicit instructions, whereas in real-world interactions, humans rarely issue instructions directly. Effective collaboration requires robots to infer user intentions proactively. In this work, we introduce *cross-modal contextual instructions, a new setting where intent is derived from spoken dialogue, environmental sounds, and visual cues rather than explicit commands.* To address this new setting, we present **RoboOmni**, a *Perceiver-Thinker-Talker-Executor* framework based on end-to-end omni-modal LLMs that unifies intention recognition, interaction confirmation, and action execution. RoboOmni fuses auditory and visual signals spatiotemporally for robust intention recognition, while supporting direct speech interaction. To address the absence of training data for proactive intention recognition in robotic manipulation, we build **OmniAction**, comprising 140k episodes, 5k+ speakers, 2.4k event sounds, 640 backgrounds, and six contextual instruction types. Experiments in simulation and real-world settings show that RoboOmni surpasses text- and ASR-based baselines in success rate, inference speed, intention recognition, and proactive assistance. We make all our datasets and code publicly available.[1]

## 1 INTRODUCTION

Vision–Language–Action (VLA) models (Zitkovich et al., 2023; Ghosh et al., 2024; Black et al., 2024) have achieved remarkable advances in robotic manipulation, leveraging large-scale cross-embodiment datasets (Padalkar et al., 2023; AgiBot-World-Contributors et al., 2025; Khazatsky et al., 2024) and Multimodal Large Language Models (MLLMs) (Wang et al., 2024; Bai et al., 2025a; Li et al., 2025). VLA models are generally categorized as (1) end-to-end models (Brohan et al., 2023; Zitkovich et al., 2023; Black et al., 2024; Kim et al., 2024; 2025), which map vision–language inputs directly to motor actions, and (2) modular Brain–Cerebellum models (Huang et al., 2023; 2024; Shi et al., 2025), which use LLMs or VLMs as planners to decompose tasks into sub-goals for low-level controllers. While modular systems emphasize explicit planning, they suffer from fragmentation and interface constraints. In contrast, end-to-end models unify vision, language, and action in a shared latent space, enabling more natural and flexible responses.

Despite notable advances in VLA research, two fundamental limitations remain. (1) From the perspective of instruction type: most works (Kim et al., 2024) focus on direct commands (Fig. 1-(a)), later extended to more complex (Fig. 1-(b)) yet explicit forms (Shi et al., 2025), while Xu et al. (2025a) recently introduced a dataset for inferential text-based instructions (Fig. 1-(c)), but system studies remain scarce. (2) From the perspective of the instruction source: current systems (Kim et al., 2024; Zitkovich et al., 2023) predominantly rely on textual instructions (Fig. 1-(d)) or ASR-transcribed speech (Fig. 1-(e)), the latter discarding essential paralinguistic cues such as

---

[†]Corresponding authors.
[1]https://github.com/OpenMOSS/RoboOmni

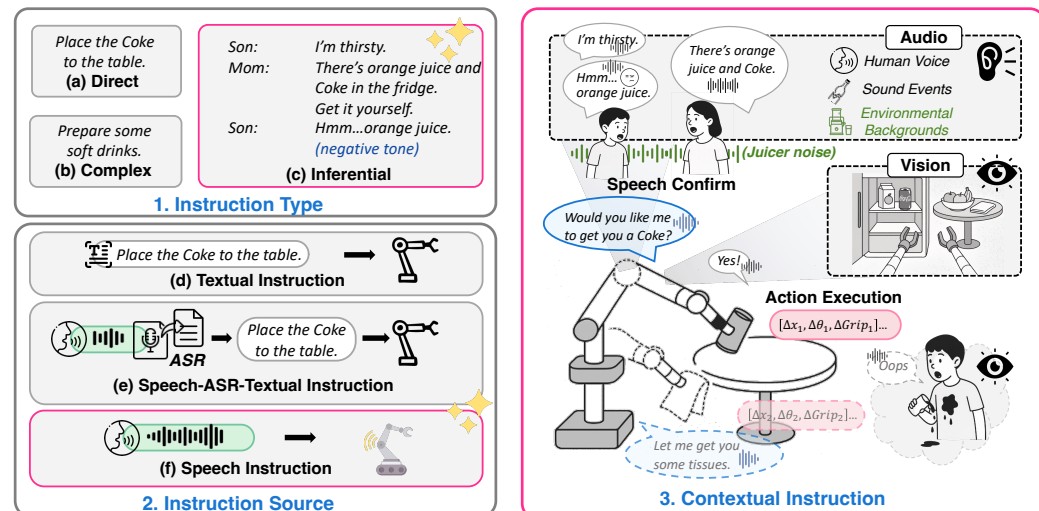

Figure 1: Overview of robotic manipulation models classified by instruction type and input. Our RoboOmni integrates contextual instruction with direct speech for end-to-end multimodal interaction and action execution.

tone, intonation, and affective signals. Recently, Zhao et al. (2025) investigated models that accept speech instructions (Fig. 1-(f)) by converting existing textual commands into speech, but neglected real-world environmental sounds. Overall, existing works assume that instructions are explicitly issued, and there is a lack of study on jointly reasoning over speech, environmental sounds, and visual observations for proactive intent recognition and reasoning.

There arises a key research question: **Can a robot integrate cross-modal context, including speech, environmental audio, and visual observations, to proactively infer and verify user intent?** As illustrated in Fig. 1, in a living-room scene, the robot integrates dialogue (audio), refrigerator observation (vision), and the juicer sound (environmental audio) to infer that John prefers cola over hand-made sour juice, and proactively seeks confirmation rather than waiting for an explicit command. Since humans rarely issue direct instructions in daily life, we define such scenarios as *cross-modal contextual instructions, where auditory (speech and environmental sound) and visual cues are fused to infer latent user intent and verified proactively by interaction*, in contrast to conventional setups that assume explicit commands.

To address these challenge and answer the research question, we propose **RoboOmni**, an end-to-end omni-modal framework for manipulation that closes the loop of intent recognition, interaction confirmation, and action execution. Unlike prior approaches, RoboOmni supports direct speech interaction without ASR, infers latent commands by fusing human speech, environmental audio, and vision through spatiotemporal modeling, and verifies intent via interaction. To overcome data scarcity, we construct OmniAction, a dataset with 140k episodes, over 5k speakers, 2.4k event sounds, 640 background sounds, and six contextual instruction types.

Experiments in both simulation and real-world settings show that RoboOmni substantially outperforms text- and ASR-based baselines, achieving higher accuracy (Sec. 5.2 and Sec. 5.3), faster inference (Sec. 5.6), more effective proactive assistance (Sec. 5.5), and improved intention recognition (Sec. 5.5). Our contributions are fourfold:

1. We introduce *cross-modal contextual instructions*, a new setting for robotic manipulation that requires robots to proactively infer user commands from multimodal context (vision, environmental sounds, and speech) rather than passively await explicit instructions.

2. We propose RoboOmni, a *Perceiver-Thinker-Talker-Executor* framework based on end-to-end omni-modal LLMs that fuses auditory and visual inputs for intent reasoning, unifying intent recognition, confirmation, and action execution.

3. To address the lack of datasets for proactive intention reasoning, we introduce OmniAction, comprising 140k episodes with 5k+ speakers, 2.4k event sounds, 640 backgrounds, and six contextual instruction types, along with OmniAction-LIBERO for simulation-based evaluation.

4. Evaluation in both simulation and real-world scenarios demonstrates that RoboOmni exhibits emerging cognitive intelligence, outperforming baselines with higher success rates, faster inference, and more effective proactive assistance and intention recognition.

## 2    RELATED WORK

The rapid development of Large Language Models (LLMs) Achiam et al. (2023); Touvron et al. (2023a) has driven progress in multimodal extensions. Multimodal LLMs (MLLMs) (202, 2023; Bai et al., 2025a; Liu et al., 2023b; Chen et al., 2023) combine text reasoning with vision, while recent end-to-end omni-modal models (Hurst et al., 2024; Xu et al., 2025b; Xie & Wu, 2024) unify speech, vision, and text but remain focused on linguistic outputs. In parallel, Vision–Language–Action (VLA) models Brohan et al. (2023); Zitkovich et al. (2023); Li et al. (2023); Team et al. (2024); Kim et al. (2024; 2025); Black et al. (2024); Li et al. (2024a); Qu et al. (2025) map instructions to actions, yet mainly assume explicit commands and struggle with context-dependent or compositional tasks. Cascaded or hierarchical variants (Intelligence et al., 2025; Shi et al., 2025; Song et al., 2025b; Lin et al., 2025) decompose goals but ignore implicit cues such as dialogue or emotion, while ASR/TTS-based speech–action pipelines (Shi et al., 2025; Khan et al., 2025) discard paralinguistic signals. Although some recent efforts handle direct speech inputs (Zhao et al., 2025), they only output actions without conversational interaction. In contrast, we propose RoboOmni, an end-to-end omni-modal framework that integrates speech, environmental sounds, vision, and text for both embodied action and natural interaction. A detailed review of related work about Omni-Modal LLMs and Vision-Language-Action Model is provided in App. B.

## 3    OMNIACTION DATASET CONSTRUCTION

### 3.1    OVERVIEW

Proactive robots must infer implicit intent from audio and visual observations, yet existing datasets lack such a combination of modalities (most of them lack audio modality) and inferential instructions needed for intent reasoning. To address this gap, we introduce OmniAction, a large-scale corpus that encodes contextual instructions—latent intents grounded in speech, environmental audio, sound events, and vision. OmniAction covers six instruction categories and three non-speech sounds.

**Diverse Contextual Instructions.** (1) *Sentiment Cues*: Emotionally tinted expressions, or subtle vocalizations, that indirectly reveal user preferences or intentions (e.g., "Ugh, this juice is too sour" implying a request for an alternative). (2) *Overlapping Voices*: Multi-speaker audio segments with temporal overlaps, testing intent extraction under crosstalk and partial masking. (3) *Non-Verbal Cues*: Salient non-linguistic audio events (e.g., alarms, phone rings) that carry situational information relevant to the task. (4) *Identity Cues*: Speaker attributes such as age and gender, inferred from voice and not available from text, are needed to decide whose intent to satisfy. (5) *Dyadic Dialogue*: Two-participants dialogues where intent emerges from conversational flow rather than explicit commands. (6) *Triadic Dialogue*: Three participants interact with turn-taking and indirect references, increasing the complexity of intent attribution. To preserve general command-following ability beyond dialogue, we also include a portion of single-person text instructions during training.

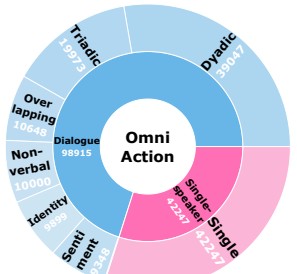

Figure 2: Distribution of contextual instruction types in OmniAction.

**Diverse Non-Speech Sounds.** We also investigate three types of acoustic variation: (1) *Speaker Timbre*. 5,096 distinct voices spanning six categories by age (elderly, adult, child) and gender (male, female). Reference audio clips are used for timbre cloning to ensure within-dialogue consistency and cross-speaker diversity. (2) *Sound Events*. 2,482 non-verbal events (e.g., thunder, doorbell) were inserted at scripted anchors to provide cues beyond speech. (3) *Environmental Backgrounds*. 640 ambient soundscapes (e.g., running water, stir-fry sizzling) mixed at controlled signal-to-noise ratios (SNRs) to mimic daily environments.

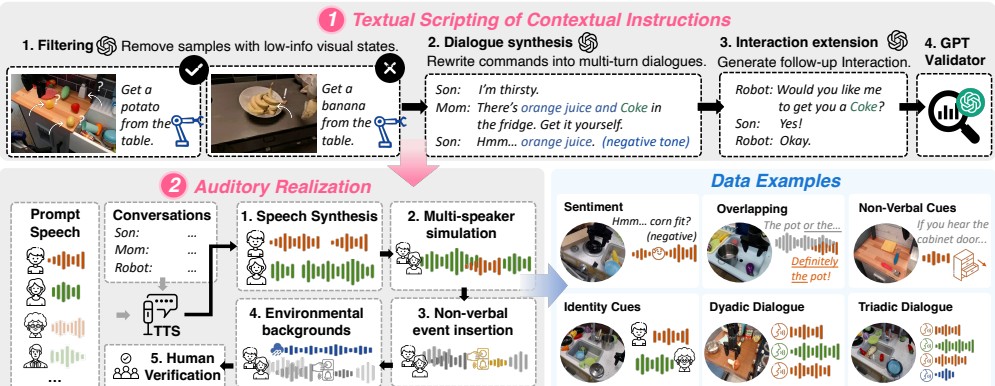

Figure 3: Overview of OmniAction Dataset Construction Pipelines and Examples.

**Data Statistics and Formats.** OmniAction comprises 141,162 multimodal episodes, spanning 112 skills (e.g., *pick-place*, *open/close*) and 748 objects (e.g., *can*), with 5,096 distinct speaker timbres, 2,482 non-verbal sound events, and 640 environmental backgrounds. Each sample is represented as a triplet $(C, V, A)$: a multi-turn conversation $C$, a visual observation sequence $V$, and an action trajectory $A = \{a_t\}_{t=1}^{T}$, where $a_t \in \mathbb{R}^7$ denotes the delta control vector of the end-effector. The distribution of instruction type is detailed in Fig. 2. More detailed statistics and examples are shown in Sec. C.1 and Sec. C.2.

## 3.2 CONSTRUCTION PROCESS

We construct OmniAction through a three-stage pipeline—textual scripting, auditory realization, and verification—illustrated in Fig. 3.

**Textual Scripting.** We sample tasks and trajectories from Open-X datasets (Padalkar et al., 2023) and transform each atomic instruction into a contextual one with GPT-4o through: (1) *Filtering*: removing trivial samples with low-information visual states. (2) *Dialogue synthesis*: rewriting instructions into multi-turn household dialogues that span six contextual instruction types. (3) *Interaction extension*: constructing follow-up human–robot exchanges that simulate natural interactions. (4) *Validation*: ensuring intent consistency with the original instruction.

**Auditory Realization.** To capture paralinguistic cues and environmental acoustics beyond text, we convert dialogues into audio that reflects real household conditions, augmented with diverse sound events and background environments. The conversion process includes four steps: (1) *Speech synthesis*: rendering user turns into audio via multiple TTS engines with voice cloning for timbre consistency and cross-dialogue diversity. (2) *Multi-speaker simulation*: generating each speaker's turns separately, concatenating them on the timeline, and inserting overlaps at controlled offsets. (3) *Non-verbal event insertion*: mixing contextual sounds (e.g., alarms, utensil clatter) at scripted anchors. (4) *Environmental backgrounds*: adding ambient textures (e.g., water flow, frying, fan hum) at varying SNRs. Further implementation details are provided in Sec. C.3.

**Verification.** To ensure data quality, we conducted a manual evaluation on sampled speech dialogues and confirmed that task intent was reliably recoverable (98.7% agreement, detailed in Sec. C.4).

## 3.3 SIMULATION DATASET: OMNIACTION-LIBERO

To address the lack of simulation benchmarks, we construct **OmniAction-LIBERO** based on LIBERO (Liu et al., 2023a), with two variants. (1) **OmniAction-LIBERO-TTS** augments the LIBERO using the pipeline described above. Starting from 40 manipulation tasks across four suites (Spatial, Object, Goal, Long-Horizon), we generate six variants for each task based on the six contextual instruction types, yielding 240 evaluation tasks. Example dialogues and task scenes are provided in App. G. (2) **OmniAction-LIBERO-Real** evaluates RoboOmni under real speech conditions, where 10 volunteers provide spoken instructions collected in real environments.

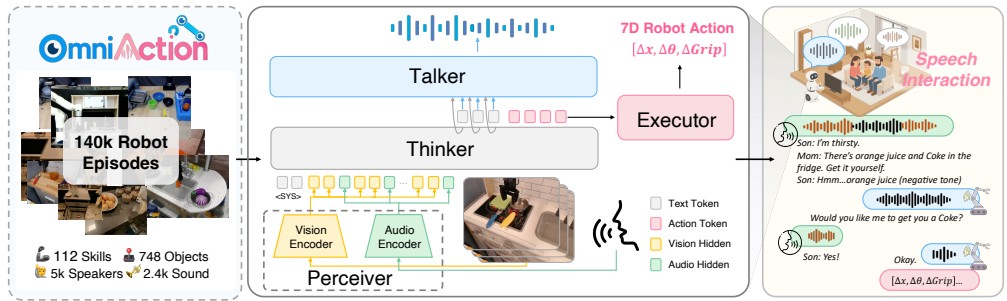

Figure 4: The framework of **RoboOmni**, a Perceiver-Thinker-Talker-Executor architecture that unifies vision, text, and audio in a shared token space to generate actions and speech.

## 4 METHODS

We propose RoboOmni, an end-to-end omni-modal LLM framework organized as Perceiver–Thinker–Talker–Executor, unifying speech, environmental audio, vision, and robotic actions within a single autoregressive model Fig. 4. RoboOmni employs a Perceiver for multimodal input, a Thinker backbone, a Talker for speech, and an Executor for actions. To align inputs with linguistic and motor outputs, RoboOmni uses unified tokenization to encode all modalities into a shared semantic space, which the Thinker processes into high-level representations and specialized decoders render into speech and executable actions, enabling seamless perception-to-action generation.

### 4.1 TASK DEFINITION

We consider *cross-modal contextual instruction following*, where the robot receives multimodal input $X = \{V_{1:T}, S_{1:T}, C\}$, consisting of visual observations $V_t$ and audio signals $S_t$ (including human speech with varying identity, prosody, emotion, overlapping voices, and environmental sounds), and dialogue history $C$. The robot must use these cues to infer latent intent, produce clarifications $y_{1:L}$ when clarification is needed, and execute actions $A_{1:N}$. Unlike standard VLA tasks, intent here is not explicitly spoken but emerges from multimodal context.

### 4.2 ARCHITECTURE COMPONENTS

**Perceiver: Multimodal Input Encoding.** The Perceiver handles the encoding of heterogeneous input modalities into a unified embedding space. Following the multimodal processing pipeline of Qwen2.5-Omni (Xu et al., 2025b), at each timestep $t$ the robot receives a visual frame $V_t$, an audio segment $S_t$, and the dialogue histroy up to that point $C_t$. Modality-specific encoders produce embeddings $\mathbf{v}_t = f_v(V_t)$, $\mathbf{s}_t = f_s(S_t)$, and $\mathbf{c}_t = f_c(C_t)$, which are combined into a unified representation $\mathbf{X}_t = [\mathbf{v}_t; \mathbf{s}_t; \mathbf{c}_t]$ that serves as input to the Thinker backbone.

**Thinker: Omni-Modal Reasoning.** The Thinker serves as the central reasoning engine, built upon the LLM backbone. It processes the unified multimodal representations from the Perceiver and generates contextually appropriate outputs in the joint vocabulary space $\mathcal{V} \cup \mathcal{A}$. The Thinker autoregressively produces sequences that seamlessly interleave text tokens, speech representations, and action tokens, enabling unified reasoning across perception, language, and robotic control.

**Talker: Speech Generation.** The Talker component enables the system to generate natural speech responses through a hierarchical architecture design. The Talker receives high-level semantic representations and text token from the Thinker and converts them into speech waveforms, allowing for seamless voice interaction in robotic scenarios.

**Executor: Action Generation.** To enable seamless integration of robotic control within the language model framework, we extend the vocabulary of the Thinker with a set $\mathcal{A}$ of 2048 discrete action tokens introduced by the FAST+ tokenizer (Pertsch et al., 2025). Rather than mapping each action dimension to a separate token, FAST+ represents a continuous action vector $a_t \in \mathbb{R}^7$ (e.g., 7-DoF control) by a short sequence of discrete symbols $r_t \subset \mathcal{A}$. This enables the model to autoregressively generate from the joint space $\mathcal{V} \cup \mathcal{A}$, where $\mathcal{V}$ represents the text vocabulary, seamlessly bridging language understanding and robotic control within a single sequence. The Executor then decodes these action tokens back into executable robot commands.

## 4.3 DUAL-MODE GENERATION

**Text and Speech Generation.** For conversational responses, the Thinker autoregressively generates text tokens $\mathbf{y}_{1:L} = (y_1, y_2, \ldots, y_L)$:

$$p_\theta(\mathbf{y}_{1:L}|\mathbf{X}_t) = \prod_{\ell=1}^{L} p_\theta(y_\ell|\mathbf{X}_t, \mathbf{y}_{<\ell}). \tag{1}$$

The generated text can optionally be converted to speech through the Talker module, which receives the discrete text tokens and high-level semantic representations from the Thinker.

**Action Generation.** For robotic control, the Thinker autoregressively predicts discrete action tokens $\mathbf{r}_{t:t+N}$ of chunk length $N$, which are decoded into continuous actions $\mathbf{a}_{t:t+N}$ by inverse transform.

$$\mathbf{a}_{t:t+N} = \text{Executor}(\mathbf{r}_{t:t+N}), \quad p_\theta(\mathbf{r}_{t:t+N}|\mathbf{X}_t) = \prod_{i=0}^{N} p_\theta(r_{t+i} \mid \mathbf{X}_t, \mathbf{r}_{t:t+i-1}). \tag{2}$$

## 4.4 TRAINING PARADIGMS

We train RoboOmni using a unified autoregressive objective that handles both conversational and manipulation capabilities within the same framework. Given a training episode, the model receives multimodal input $\mathbf{X}_t$ and learns to predict appropriate responses—either conversational replies for dialogue or action sequences for manipulation.

For conversational interactions, the model optimizes the likelihood of generating appropriate text responses $\mathbf{y}_{1:L}$ given the multimodal context:

$$\mathcal{L}_{\text{chat}}(\theta) = -\mathbb{E} \sum_{\ell=1}^{L} \log p_\theta(y_\ell|\mathbf{X}_t, \mathbf{y}_{<\ell}). \tag{3}$$

For action generation, the model learns to generate action token sequences $\mathbf{r}_{t:t+N}$ that correspond to expert trajectory:

$$\mathcal{L}_{\text{act}}(\theta) = -\mathbb{E} \sum_{i=0}^{N} \log p_\theta(r_{t+i}|\mathbf{X}_t, \mathbf{r}_{t:t+i-1}). \tag{4}$$

The complete training objective combines both modalities through batch interleaving:

$$\mathcal{L}(\theta) = \mathcal{L}_{\text{chat}}(\theta) + \mathcal{L}_{\text{act}}(\theta) = -\mathbb{E} \sum_{k=1}^{K} \log p_\theta(z_k \mid \mathbf{X}_t, z_{<k}), \quad z_k \in \mathcal{V} \cup \mathcal{A}, \tag{5}$$

which highlights that both conversational and action supervision reduce to the same autoregressive maximum-likelihood objective over a unified token space.

## 5 EXPERIMENT

### 5.1 EXPERIMENT SETUP

**Baseline Models** As current open-source Vision-Language-Action (VLA) models are primarily designed for textual instructions and cannot directly process audio inputs, we construct two baseline paradigms to validate the necessity of end-to-end audio processing: (i) **Ground-truth Textual Prompt**, which directly feeds pre-annotated transcriptions of speech instructions into VLA models; (ii) **Speech-ASR-Textual Prompt**, where speech instructions are first transcribed to text using the ASR model Whisper large-v3 (Radford et al., 2023), then fed into VLA models. We conduct evaluations comparing RoboOmni with four representative VLA baselines representing both paradigms: **OpenVLA** (Kim et al., 2024), **OpenVLA-OFT** (Kim et al., 2025), $\pi_0$ (Black et al., 2024), and **NORA** (Hung et al., 2025). Details of these baselines are in App. D.

**Implementation Details** We train the model with an input image resolution of $224 \times 224$, an audio sampling rate of 16,000 Hz, and an action chunk size of 6. For large-scale pretraining, RoboOmni is optimized on a cluster of 64 A100 GPUs over 10 days, corresponding to a total of 15,360 A100-hours, with a batch size of 512. The training runs for 10 epochs using a learning rate of $5 \times 10^{-5}$, with the first 1k steps reserved for warm-up. For downstream task supervised fine-tuning (SFT), we adopt a learning rate of $5 \times 10^{-5}$ and train with 8 A100 GPUs for 10-30k steps.

Table 1: Performance of different robot manipulation models on the OmniAction-LIBERO-TTS benchmark, evaluated across four task suites (Spatial, Goal, Object, Long-Horizon) under six contextual instruction types. Values in **bold** denote the best performance.

| | Task | Ground-truth Textual Prompt | | | | Audio → ASR → Text Prompt | | | | RoboOmni |
|---|---|---|---|---|---|---|---|---|---|---|
| | | OpenVLA | OFT | NORA | $\pi_0$ | OpenVLA | OFT | NORA | $\pi_0$ | |
| Spatial | *Sentiment* | 4.0 | 9.0 | 40.0 | 8.0 | 1.0 | 8.0 | 43.0 | 11.0 | **93.0** |
| | *Non-Verbal* | 2.0 | 8.0 | 61.0 | 7.0 | 3.0 | 8.0 | 68.0 | 14.0 | **91.0** |
| | *Identity* | 1.0 | 8.0 | 53.0 | 4.0 | 2.0 | 18.0 | 56.0 | 7.0 | **92.0** |
| | *Overlapping* | 6.0 | 7.0 | 43.0 | 7.0 | 11.0 | 6.0 | 58.0 | 18.0 | **93.0** |
| | *Dyadic* | 7.0 | 6.0 | 51.0 | 5.0 | 4.0 | 17.0 | 57.0 | 3.0 | **95.0** |
| | *Triadic* | 1.0 | 7.0 | 51.0 | 6.0 | 2.0 | 6.0 | 57.0 | 6.0 | **94.0** |
| | Avg | 3.5 | 7.5 | 49.8 | 6.2 | 3.8 | 10.5 | 56.5 | 9.8 | **93.0** |
| Goal | *Sentiment* | 0.0 | 0.0 | 11.0 | 0.0 | 0.0 | 0.0 | 9.0 | 3.0 | **89.0** |
| | *Non-Verbal* | 0.0 | 0.0 | 18.0 | 0.0 | 1.0 | 0.0 | 22.0 | 4.0 | **79.0** |
| | *Identity* | 0.0 | 0.0 | 11.0 | 3.0 | 0.0 | 0.0 | 11.0 | 1.0 | **82.0** |
| | *Overlapping* | 0.0 | 0.0 | 21.0 | 0.0 | 0.0 | 0.0 | 23.0 | 1.0 | **97.0** |
| | *Dyadic* | 0.0 | 0.0 | 7.0 | 1.0 | 1.0 | 10.0 | 18.0 | 0.0 | **85.0** |
| | *Triadic* | 0.0 | 0.0 | 7.0 | 2.0 | 0.0 | 0.0 | 15.0 | 0.0 | **83.0** |
| | Avg | 0.0 | 0.0 | 12.5 | 1.0 | 0.3 | 1.7 | 16.3 | 1.5 | **85.8** |
| Object | *Sentiment* | 1.0 | 0.0 | 9.0 | 4.0 | 2.0 | 0.0 | 5.0 | 6.0 | **83.0** |
| | *Non-Verbal* | 2.0 | 0.0 | 7.0 | 1.0 | 3.0 | 0.0 | 17.0 | 8.0 | **82.0** |
| | *Identity* | 4.0 | 0.0 | 4.0 | 5.0 | 5.0 | 0.0 | 15.0 | 8.0 | **85.0** |
| | *Overlapping* | 14.0 | 7.0 | 1.0 | 6.0 | 26.0 | 0.0 | 16.0 | 9.0 | **84.0** |
| | *Dyadic* | 20.0 | 0.0 | 14.0 | 7.0 | 20.0 | 10.0 | 19.0 | 7.0 | **88.0** |
| | *Triadic* | 2.0 | 0.0 | 3.0 | 5.0 | 2.0 | 10.0 | 11.0 | 2.0 | **82.0** |
| | Avg | 7.2 | 1.2 | 6.3 | 4.7 | 9.7 | 3.3 | 13.8 | 6.7 | **84.0** |
| Long | *Sentiment* | 0.0 | 0.0 | 26.0 | 4.0 | 0.0 | 0.0 | 50.0 | 5.0 | **76.0** |
| | *Non-Verbal* | 0.0 | 0.0 | 35.0 | 1.0 | 0.0 | 0.0 | 57.0 | 2.0 | **76.0** |
| | *Identity* | 0.0 | 0.0 | 29.0 | 4.0 | 1.0 | 0.0 | 43.0 | 4.0 | **79.0** |
| | *Overlapping* | 0.0 | 0.0 | 35.0 | 5.0 | 3.0 | 0.0 | 56.0 | 6.0 | **79.0** |
| | *Dyadic* | 1.0 | 0.0 | 42.0 | 1.0 | 1.0 | 0.0 | 59.0 | 5.0 | **85.0** |
| | *Triadic* | 0.0 | 0.0 | 27.0 | 5.0 | 2.0 | 10.0 | 41.0 | 8.0 | **82.0** |
| | Avg | 0.2 | 0.0 | 32.3 | 3.3 | 1.2 | 1.7 | 51.0 | 5.0 | **79.5** |
| Avg | | 2.6 | 0.4 | 16.3 | 3.0 | 3.9 | 2.3 | 25.9 | 4.4 | **85.6** |

## 5.2 EVALUATION ON CROSS-MODAL CONTEXTUAL INSTRUCTIONS

To comprehensively evaluate RoboOmni on diverse cross-modal contextual instructions, we conduct extensive experiments on the OmniAction-LIBERO across four task suites with six audio variants. Tab. 1 demonstrates that RoboOmni achieves an overall 85.6% success rate, substantially outperforming the strongest baseline (NORA, 25.9%) and other cascaded methods (all below 10%). Our analysis yields three key insights: (1) **End-to-end auditory integration is crucial for paralinguistic cues.** Text-only models, whether using ASR transcripts or ground-truth text, fail to capture paralinguistic cues (e.g., prosody, overlapping speech), with best scores of 25.9% (textual baseline). In contrast, RoboOmni's direct audio processing enables it to consistently exceed 76% across all types, demonstrating the importance of preserving acoustic information. (2) **Auditory integration enhances robust intent recognition under ambiguity.** Goal and Object suites are challenging due to multiple manipulable objects and valid actions, where baselines collapse (averaging 16.3% and 13.8% for the best baselines), exposing limits in contextual instruction understanding. RoboOmni sustains high performance (Goal: 85.8% v.s. Object: 84.0%), demonstrating robust generalization under semantic ambiguity. (3) **Instruction type complexity reveals varying cognitive demands.** For end-to-end models, *dyadic* and *overlapping* tasks are easier, averaging ∼88%. *Non-verbal* instructions are hardest (∼82%), as they require recognizing non-verbal sounds and integrating them with visual and speech cues. The remaining tasks average ∼85%, reflecting moderate complexity.

## 5.3 EVALUATION ON REAL HUMAN AUDIO DIRECT INSTRUCTIONS

We further evaluate RoboOmni's robustness under real human-recorded speech with direct audio instructions. As shown in Tab. 2, on the OmniAction-LIBERO-Real benchmark, RoboOmni achieves the highest average performance (76.6%), surpassing strong text-based VLAs including $\pi_0$ (73.8%), OpenVLA (40.1%), and NORA (17.4%). ASR-based VLAs suffer from acoustic variability:

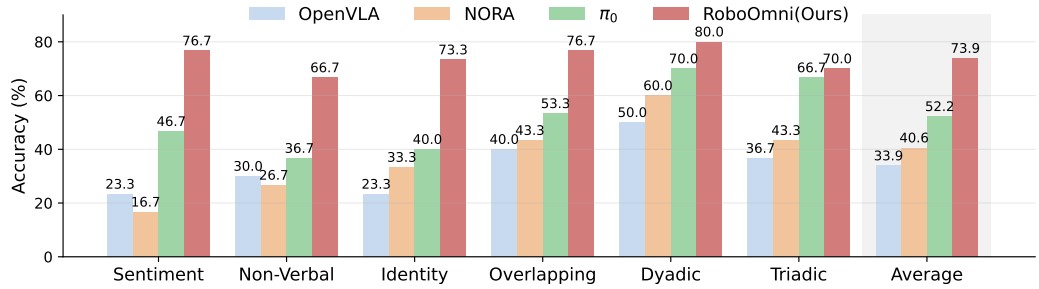

Figure 5: Real-world contextual-instruction performance with human speech. RoboOmni surpasses ASR+VLA baselines by directly grounding raw audio and vision and retaining paralinguistic cues.

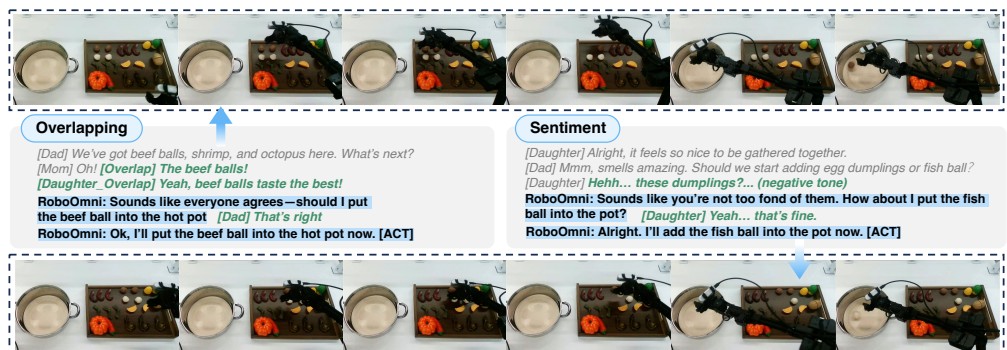

Figure 6: Demonstration of success cases of RoboOmni on the real-world WidowX 250S robot arm.

accents, coarticulation, and background noise frequently cause recognition errors, and even minor word deviations can degrade VLAs' performance. $\pi_0$ shows some robustness, likely due to large-scale co-training on diverse web data. In contrast, RoboOmni processes speech directly, avoiding ASR pipeline errors. Pretraining on diverse speakers and sounds improves robustness to acoustic variability and paralinguistic cues, yielding more consistent performance.

Table 2: Performance comparison on OmniAction-LIBERO-Real.

| | Spatial | Goal | Object | Long | Avg |
|---|---|---|---|---|---|
| Audio → ASR → Text Prompt | | | | | |
| OpenVLA | 51.6 | 38.2 | 38.0 | 32.4 | 40.1 |
| OpenVLA-OFT | 6.6 | 9.8 | 9.8 | 0.0 | 6.5 |
| NORA | 2.0 | 5.6 | 26.8 | 35.4 | 17.4 |
| $\pi_0$ | 86.0 | 60.0 | 70.0 | **79.0** | 73.8 |
| Ours (Audio Input) | | | | | |
| RoboOmni | **89.0** | **71.6** | **75.1** | 75.0 | **76.6** |

## 5.4 REAL-WORLD EXPERIMENTS

To verify that RoboOmni's capabilities transfer beyond simulation, we fine-tune our pretrained model by utilizing our demonstration dataset on WidowX 250S, where speech was recorded by 10 volunteers in real environments. This enables RoboOmni to run on real robots and handle diverse speech instructions (e.g., sentiment, overlapping cues). We compare RoboOmni against several ASR+VLA baselines trained with the same data. Each task is executed for 10 trials, and we report the mean task success rate. As shown in Fig. 5, RoboOmni achieves 73.9% success, substantially outperforming the best ASR+VLA baseline (52.2%). This performance gain primarily comes from two advantages of the unified architecture: (1) RoboOmni directly reasons over raw audio and vision, making it robust to natural speech variations, whereas cascaded systems are prone to ASR errors and to VLA brittleness under small transcription changes. (2) The end-to-end multimodal design preserves paralinguistic cues—sentiment, speaker identity, and non-verbal events—essential for contextual-intent inference, while ASR pipelines inevitably discard them.

Fig. 6 highlights RoboOmni's real-world competence across three dimensions: (1) strong intent recognition, accurately inferring user intention from both visual and auditory cues (e.g., identifying the object based on audio and determining the receptacle is the pot from the visual scene); (2) effective interaction, proactively asking clarifying questions after inferring the user's latent intent (e.g., "should I . . . ?") and executing the action after receiving confirmation; (3) reliable execution, successfully carrying out the confirmed action. More detailed cases are provided in Sec. E.1

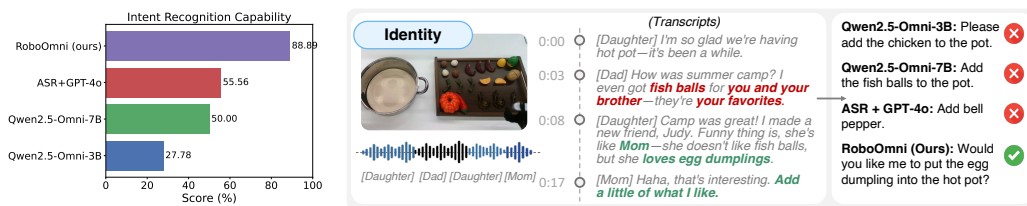

(a) Intent recognition capability        (b) Qualitative comparison of interaction capabilities.

Figure 7: Qualitative and quantitative Evaluation of Proactive Assistance Capabilities.

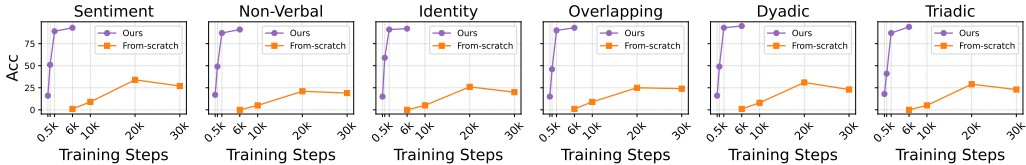

Figure 8: Training efficiency comparison between OmniAction-pretrained + SFT vs. from-scratch SFT on OmniAction-LIBERO (Spatial).

## 5.5 EVALUATION OF PROACTIVE ASSISTANCE CAPABILITIES

**Intent Recognition Capability.** We further evaluate the models' ability to recognize user intent under contextual instructions, shown in Fig. 7a. Specifically, we compare Qwen2.5-Omni-3B (our backbone), Qwen2.5-Omni-7B, and ASR+GPT-4o, against our proposed RoboOmni. We observe that RoboOmni achieves the highest accuracy (88.9%), confirming the advantage of end-to-end speech–action modeling that preserves paralinguistic cues and dialogue context. Notably, although ASR introduces recognition noise compared with end-to-end models, GPT-4o still surpasses the smaller Omni models (55.6% vs. 27.8%/50.0%) because its stronger multimodal reasoning compensates for transcription loss. This highlights that contextual instructions cannot be resolved by acoustic modeling alone, but also demand robust reasoning capabilities

**Interaction Capability.** We qualitatively assess models' interaction capability in handling contextual instructions. As shown in Fig. 7b, RoboOmni excels by proactively clarifying, integrating cross-modal signals, and sustaining natural dialogue, whereas baselines often fail in one or more aspects. Additional case studies for all instruction types appear in Sec. E.2.

## 5.6 FURTHER ANALYSIS

**What components of the input drive RoboOmni's gains?** To understand which multimodal inputs drive RoboOmni's performance, we conduct input-controlled ablations on the intent-recognition experiment, keeping the architecture fixed while specific inputs are removed: (1) w/o vision, removing visual input; (2) w/o audio, removing the audio; (3) w/o paralinguistic cues, where audio is re-recorded by a single neutral speaker without prosody, emotion, or non-verbal events; and (4) Full Input (ours). As shown in Tab. 3, Full Input achieves 88.89%, while performance drops to 58.89% without vision, 50.56% without paralinguistic cues, and 11.11% without audio. These results reveal that: (1) Audio provides the core semantic instruction, as removing it eliminates actionable content; (2) Vision is essential for contextual grounding, especially when tasks involve spatial relations or object attributes; (3) Paralinguistic cues significantly aid disambiguation across identity-, emotion-, and sound event-dependent cases. Overall, RoboOmni's gains stem from the complementary integration of vision, speech semantics, and paralinguistic signals, underscoring the need for unified end-to-end multimodal modeling.

Table 3: Ablation study on intent recognition under different configurations.

| Setting | Accuracy (%) |
| --- | --- |
| Full Input (ours) | 88.89 |
| w/o vision | 58.89 |
| w/o audio | 11.11 |
| w/o paralinguistics | 50.56 |

**Does OmniAction Pretraining Improve Training Efficiency?** To evaluate the benefit of pretraining on OmniAction, we compare finetuning efficiency on the six Spatial variants in OmniAction-

LIBERO, contrasting OmniAction-pretrained + SFT with from-scratch SFT (Fig. 8). The pre-trained model converges rapidly, reaching nearly 90% accuracy within 2k steps, while the from-scratch counterpart only attains ∼30% after 20k steps and even degrades at 30k steps. This highlights that pre-training on OmniAction providing strong generalizable priors for fast and stable adaptation with minimal fine-tuning.

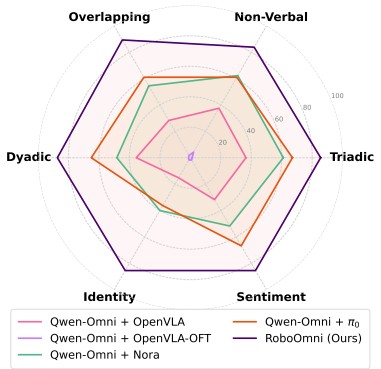

**Can Cascaded Pipelines Handle Contextual Instructions Effectively with High-level Planner?** We compare RoboOmni with planner–controller pipelines, where Qwen2.5-Omni-3B serves as the planner and text-based VLAs as controllers, shown in Fig. 9, evaluated on the OmniAction–LIBERO benchmark. RoboOmni outperforms all cascaded pipelines, demonstrating the advantage of end-to-end speech–action learning: jointly modeling audio, vision, and action avoids the lossy planner–controller interface and preserves intent fidelity. Cascaded pipelines perform worse due to (1) semantic drift, as planners are not co-trained with VLAs and generate commands controllers cannot execute, and (2) poor handling of speaker identity, since Qwen-Omni fails to capture paralinguistic cues, leading to the weakest results on *Identity Cues*.

Figure 9: Comparison between end-to-end RoboOmni and cascaded planner–controller pipelines across six contextual instruction types.

**Does End-to-End Modeling Improve Inference Efficiency?** To assess whether end-to-end modeling improves runtime efficiency, we measure per-inference latency on a single RTX 4090 GPU. Using ASR + OpenVLA as the baseline (1.0×), we find that other cascaded pipelines (ASR + Nora: 1.02×, ASR + $\pi_0$: 0.96×) incur similar costs since the ASR stage dominates computation. In contrast, RoboOmni runs at 0.49× latency, showing that end-to-end audio–action modeling eliminates the ASR bottleneck and substantially improves efficiency ( Fig. 10).

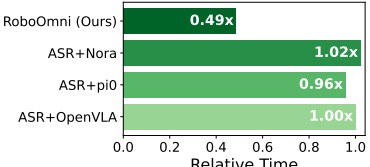

Figure 10: Per-inference latency comparing cascaded pipelines and RoboOmni.

**Failure Analysis** A task is counted as successful only when robots both *infers the correct intent* and *executes the action*. To understand where failures occur, we categorize all error cases of real-world experiment into nine interpretable types spanning intention-level and execution-level errors. As shown in Fig. 11, 42.6% intention-related failures mainly come from identity attribution errors, sentiment cue misreading. The remaining 57.4% cases arise from execution issues, dominated by grasp failures, followed by pose estimation drift and reachability constraints. This breakdown clarifies RoboOmni's current bottlenecks and highlights where future improvements in audio grounding and low-level control are most needed.

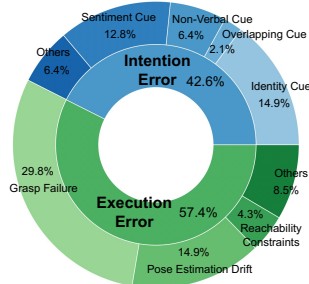

Figure 11: Failure analysis of real-world experiments.

## 6 CONCLUSION

In conclusion, we introduced cross-modal contextual instructions, a new paradigm for robotic manipulation where robots proactively infer user intent from multimodal context—vision, environmental sounds, and speech—rather than passively awaiting explicit commands. Building on this setting, we proposed RoboOmni, a Perceiver–Thinker–Talker–Executor framework built on end-to-end omni-modal LLMs that integrates auditory and visual inputs, unifying intention recognition, confirmation, and action execution. To address data scarcity, we constructed OmniAction, a large-scale corpus of 140k episodes with diverse speakers, event sounds, and backgrounds, together with OmniAction-LIBERO for simulation-based evaluation. Comprehensive experiments in both simulation and the real world demonstrate that RoboOmni exhibits emerging cognitive intelligence, significantly outperforming text- and ASR-based baselines in success rate, inference speed, proactive assistance, and intention recognition.

ACKNOWLEDGMENTS

This work was supported by the National Natural Science Foundation of China (No. 62521004). This research/project is supported by the National Research Foundation, Singapore under its National Large Language Models Funding Initiative (AISG Award No: AISG-NMLP-2024-002). Any opinions, findings and conclusions or recommendations expressed in this material are those of the author(s) and do not reflect the views of National Research Foundation, Singapore.

ETHICS STATEMENT

This work equips robots with extended contextual instructions, including family speech dialogues, environmental sounds, and visual observations, to enable more natural multimodal interaction. While such data enhances robotic intelligence, we recognize the importance of protecting user privacy when scaling to broader deployments. All collected dialogue and audio data are restricted to academic research use only and will not be shared for other purposes. Future applications must carefully manage privacy, consent, and secure handling of user interactions to ensure compliance with ethical and legal standards.

REPRODUCIBILITY STATEMENT

We will open-source the OmniAction dataset, model checkpoints, and training code to facilitate further research in this field. Training details are described in Sec. 5.1, and the complete OmniAction data construction process is documented in Sec. 3 and App. C.

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

## A LLM USAGE

Large Language Models (LLMs) were used to aid in the writing and polishing of the manuscript. Specifically, we used an LLM to assist in refining the language, improving readability, and ensuring clarity in various sections of the paper. The model helped with tasks such as sentence rephrasing, grammar checking, and enhancing the overall flow of the text.

It is important to note that the LLM was not involved in the ideation, research methodology, or experimental design. All research concepts, ideas, and analyses were developed and conducted by the authors. The contributions of the LLM were solely focused on improving the linguistic quality of the paper, with no involvement in the scientific content or data analysis.

The authors take full responsibility for the content of the manuscript, including any text generated or polished by the LLM. We have ensured that the LLM-generated text adheres to ethical guidelines and does not contribute to plagiarism or scientific misconduct.

## B DETAILED RELATED WORK

**Omni-Modal LLMs** The rapid development of Large Language Models (LLMs) Achiam et al. (2023); Touvron et al. (2023a) has spurred progress in multimodal extensions. Multimodal LLMs (MLLMs) (202, 2023; Bai et al., 2025a; Liu et al., 2023b; Chen et al., 2023) augment text-based reasoning with visual perception, enabling instruction following grounded in images. Early attempts toward omni-modality relied on modular pipelines that separately process speech and vision (Wu et al., 2023; Zhan et al., 2024; Lu et al., 2023), which makes temporal alignment across modalities difficult and limits accurate understanding of situated semantics. More recent work has shifted toward end-to-end omni-modal models (Hurst et al., 2024; Xu et al., 2025b; Xie & Wu, 2024), which can jointly model speech, vision, and text in a unified representation. However, these models remain oriented toward linguistic outputs (text or audio) and do not generate embodied actions, restricting their applicability in robotics. In contrast, our work brings omni-modality into the embodied domain by introducing RoboOmni, an end-to-end framework that integrates speech, environmental sounds, visual context, and text for both action execution and proactive human–robot interaction.

**Vision-Language-Action Model** Recent studies have explored the application of large Vision–Language Models (VLMs) in robotics (Ma et al., 2024; Zhong et al., 2025), leveraging their ability to align linguistic instructions with visual scenes. Building on large-scale demonstrations, recent works develop end-to-end Vision–Language–Action (VLA) models that map vision and language to actions Brohan et al. (2023); Zitkovich et al. (2023); Li et al. (2023); Team et al. (2024); Kim et al. (2024; 2025); Black et al. (2024); Li et al. (2024a); Qu et al. (2025), but these typically assume short, explicit commands and fail on compositional or context-dependent tasks. Cascaded or hierarchical extensions (Intelligence et al., 2025; Shi et al., 2025; Song et al., 2025b; Lin et al., 2025; Song et al., 2025a) decompose instructions into sub-goals, yet remain fragmented and rigid, and neither paradigm captures *contextual instructions*—implicit intent conveyed by dialogue, tone, or visual context, which is common in human–robot interaction.

Additionally, most prior studies further treat text as the main channel, using ASR/TTS cascades to bridge speech and action (Shi et al., 2025; Khan et al., 2025; Li et al., 2024b). Such pipelines discard paralinguistic cues (e.g., emotion, speaker identity), add latency, and disrupt temporal alignment with vision. Some works incorporate environmental sounds as an additional modality (Yamakawa et al., 2011; Zhao et al., 2023; Liu et al., 2024; Jones et al., 2025), but they do not model spoken instructions or conversational intent. A few recent efforts (Zhao et al., 2025) extend VLAs to handle direct speech-based commands, yet these remain restricted to atomic or complex speech instructions and can only output actions, without the ability to respond through speech. In contrast, our work introduces an end-to-end omni-modal framework that directly integrates speech, environmental sounds, vision, and text, enabling both action execution and cross-modal contextual instruction following for natural human–robot interaction.

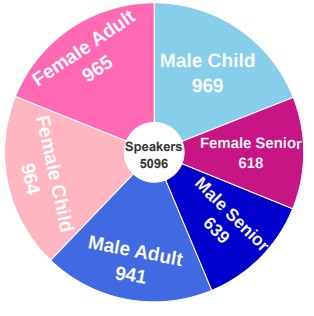

(a) Distribution of speaker timbres across six demographic categories.

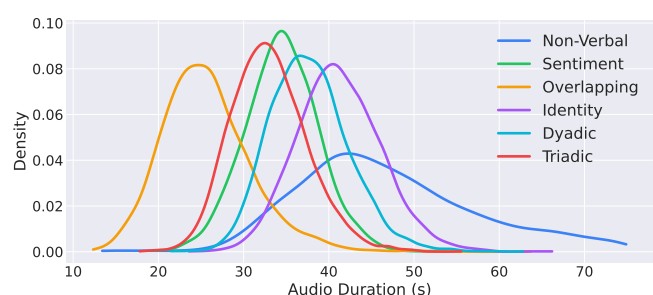

(b) Distribution of audio segment lengths across contextual instruction types.

Figure 12: Speaker and audio segment lengths statistics in OmniAction..

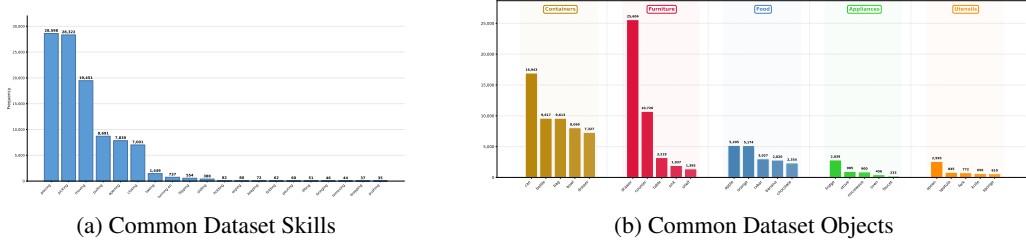

(a) Common Dataset Skills

(b) Common Dataset Objects

Figure 13: The OmniAction dataset contains a great diversity of skills and target objects.

## C  DETAILS OF OMNIACTION

### C.1  DATA STATICS

From the Open-X dataset, we filter out a subset of **74,645 base trajectories**, which are then expanded into **141,162 multimodal episodes**.

To closely approximate real conversational scenarios, **OmniAction** incorporates a diverse set of speakers covering **5,096 distinct timbres**. These span six demographic categories: male senior, female senior, male adult, female adult, male child, and female child. Fig. 12a illustrates the overall distribution of speaker timbres.

In terms of contextual instruction, Fig. 12b presents the distribution of audio segment lengths across different types of instructions. The majority of clips range from **10 to 80 seconds**. Overlapping dialogues tend to be shorter in duration, while non-verbal sequences are longer due to the insertion of sound events.

On the action-execution side, we applied natural language processing tools to the **70k trajectories** in OmniAction and extracted verb–noun pairs from the original instructions. This yields a vocabulary of **112 unique skills** and **748 manipulable objects**, as summarized in Fig. 13.

### C.2  DATA EXAMPLE

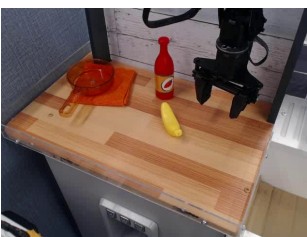

**Audio Type**: Non-Verbal Cues

**Original Instruction**:
Move the pan to the right of the bottle and banana.

**Conversation (Transcripts)**:
Daughter: Dad, can you help me with this? I'm not sure where to put the pan.
Dad: Sure, where do you think it should go?
Daughter: I think it should be moved, but I'm not sure where.
Dad: If you hear the ceramic collision, move the pan to the right of the bottle and banana. If you hear the door knob noise, leave it where it is.
Daughter: Got it. Let's see what happens.
Robot: "Should I move the orange pan to the right of the bottle and banana on the countertop?"
Dad; Yes, that's exactly what I want you to do right now, thank you!
Robot: Alright, I'll move the orange pan to the right of the bottle and banana. [ACT]

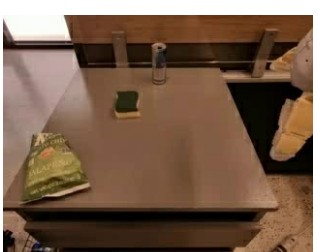

**Audio Type**: Sentiment Cues

**Original Instruction**:
put knife on cutting board

**Conversation (Transcripts)**:
Mom: Hmm, there are a few interesting items here. We could pick up that green chip bag, the sponge, or even the Red Bull can. What do you think we should go for first?
Son: Hmm... I don't think the sponge is what we're looking for—it's more for cleaning, right?
Mom Ah, not quite sure about the Red Bull either... It's small and doesn't seem to fit with what we need now.
Son: Let me think... The chip bag looks like the most likely option here.
Mom: Hmm, I see what you mean. Alright, let's go with that one.
Robot: Should I pick up the green jalapeno chip bag for you?
Mom: Yes, please pick up the green jalapeno chip bag.
Robot: OK, I will do that. [ACT]

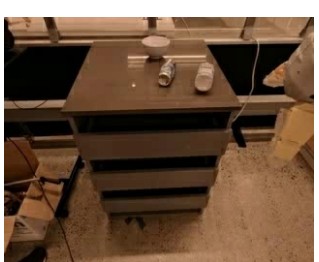

**Audio Type**: Identity Cues

**Original Instruction**:
place the silver pot on the yellow napkin

**Conversation (Transcripts)**:
Grandpa: Ah, it's good to see everything organized here. I know my little grandson loves fiddling with things, especially the top drawer. And your dad, well, he keeps saying he needs his tools from the bottom one.
Father: Haha, you know me too well, Dad. The bottom drawer has all my essentials, but I guess I could wait.
Grandpa:Well, patience runs in the family, doesn't it?
Son: Grandpa, can I check the top drawer first? Please? It's my favorite spot!
Robot: Do you need assistance to open the top drawer?
Father: Uh, yeah, sure, that'd be great!,
Robot: Alright. I will open the top drawer for you. [ACT]

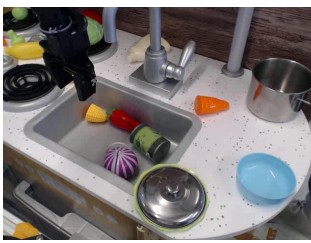

**Audio Type**: Overlapping Cues

**Original Instruction**:
put lid on pot or pan

**Conversation (Transcripts)**:
Mother: Hey, did you see the lid sitting over there?
Son: Yeah, should we use it for the pot?
Mother: I was thinking about covering the pan instead [Overlap] wait, the pot?
[Overlap_Son] Yes, the pot! It's right there.
Mother: Oh, okay, you're right. Let's cover the pot then.
Son: Haha, I knew you'd agree!
Robot: Do you need me to put the lid on the pot?
Mother: Yes, please put the lid on the pot.
Robot: OK, I will do that. [ACT]

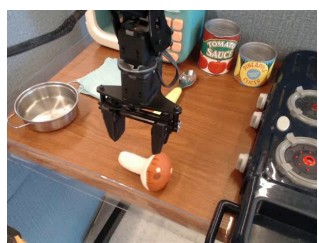

**Audio Type**: Dyadic Dialogue

**Original Instruction**:
Move pot onto the towel

**Conversation (Transcripts)**:
Dad: Oh, look at that pot sitting there.
Mom: Yeah, it's right next to the towel.
Dad: Hmm, if we move it onto the towel, it'll be easier to clean later.
Mom: Good idea, let me handle that.
Robot: Do you need me to move the pot onto the towel?
Mom: Uh, yeah, that'd be great. It's better to keep it there to avoid scratches.
Robot: Alright, I will move the pot onto the towel now. [ACT]

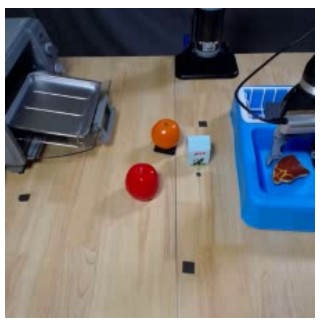

**Audio Type**: Triadic Dialogue

**Original Instruction**:
Place the brush on top of orange cloth

**Conversation (Transcripts)**:
Son: Hey, Lily, look at that steak meat over there!
Daughter: Oh, yeah! It's just sitting there like it's waiting for a prince to rescue it.
Grandpa: You two are hilarious. I guess the 'prince' could be me, huh?
Son: Go for it, Grandpa. You've got that royal touch.
Grandpa: Alright, kids, I'll pick it up and get it ready for dinner. Can't let a good steak go to waste!,
Robot: Hey, do you need me to pick up the red ball for you?
Daughter: Uh, no, we don't need that right now!
Robot: Alright, how about I pick up the steak meat instead?
Daughter: Yeah, that's what we're looking for!
Robot: OK. I will perform operation pick up the steak meat now.[ACT]

## C.3 AUDIO PERCEPTUAL INSTRUCTION GENERATION

To ensure reproducibility and clarity, we detail the full pipeline for audio perceptual instruction construction:

- Speech synthesis. We employ three complementary TTS engines—MOSS-TTSD (Open-MOSS, 2025), CosyVoice (Du et al., 2025), and Gemini-TTS[2]. Short reference clips are used for speaker voice cloning, ensuring timbre consistency across turns within a dialogue while preserving diversity across samples.

- Multi-speaker simulation. Dialogues are synthesized by rendering each speaker's utterances individually and concatenating them on a global timeline. Overlapping speech is introduced by inserting secondary utterances at controlled temporal offsets. Alignment is managed using CTC-based methods (Graves et al., 2006), enabling realistic crosstalk and interruption.

- Non-verbal events. A curated library of 2,482 non-verbal sounds (e.g., phone rings, utensil clatter, alarms, door knocks) is employed. Events are inserted at pre-defined anchors linked to dialogue semantics (e.g., placing a cup, answering a call).

- Environmental backgrounds. We sample 640 ambient textures covering diverse household conditions (e.g., running water, frying, fan hum). Each texture is mixed with the dialogue at a randomly chosen signal-to-noise ratio (SNR), spanning a wide range to simulate varying acoustic difficulty.

This augmentation pipeline provides both paralinguistic variation (speaker identity, overlap, vocal timbre) and environmental realism (non-verbal sounds, ambient noise), yielding training data that closely reflects natural household interactions.

## C.4 DETAILS OF VERFICATION

**Annotation Guidelines** For the human verification study, annotators were instructed to evaluate each dialogue–operation pair along two primary dimensions:

---

[2]https://cloud.google.com/text-to-speech/docs/gemini-tts

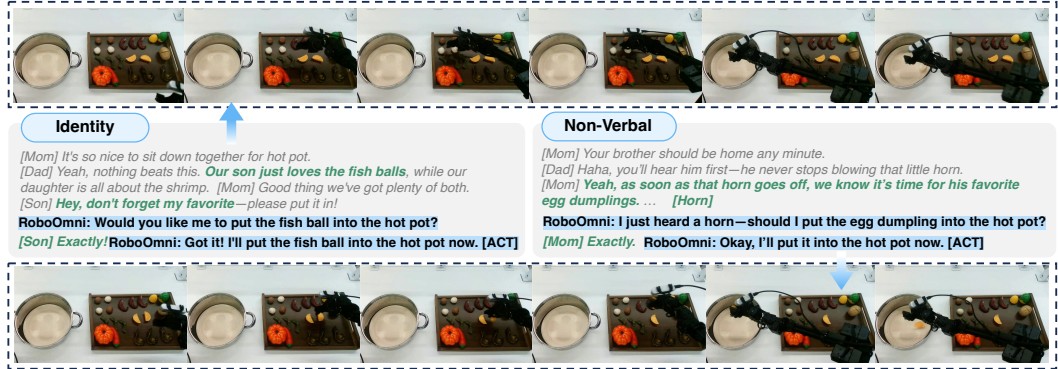

Figure 14: Demonstration of success cases of RoboOmni on the *Identity Cues* and *Non-verbal*.

1. **Intent recoverability:** Whether the latent task intent (i.e., the canonical atomic instruction) can be unambiguously inferred from the dialogue and multimodal context. Annotators were asked to answer *Yes/No*, with *Yes* requiring that the original intent could be reasonably reconstructed without external information.

2. **Phenomenon fidelity:** Whether the dialogue faithfully realizes the targeted phenomenon category. Examples include:

   - *Sentiment:* successful inference of intent requires recognizing sentiment-laden cues (e.g., dislike, refusal).
   - *Overlapping:* the audio contains genuine temporal overlaps such that ASR alone would be challenged.
   - *Non-Verbal:* correct inference depends on a salient non-verbal sound event (e.g., alarm, phone ring).
   - *Identity:* the requesting agent must be distinguishable via age/gender/role cues.
   - *Dyadic / Triadic:* the task intent is embedded within multi-turn, two- or three-party exchanges.

Annotators were provided with ten positive and ten negative examples per phenomenon before annotation began, serving as calibration.

## D  DETAILS OF BASELINE

We compare against four representative VLA baselines: (1) **OpenVLA** (Kim et al., 2024), built on Llama-2 (Touvron et al., 2023b) with DINOv2 (Oquab et al., 2023) and SigLIP (Zhai et al., 2023) encoders, pretrained on ∼970k demonstrations from Open-X-Embodiment (Padalkar et al., 2023). (2) **OpenVLA-OFT** (Kim et al., 2025), a variant of OpenVLA augmented with action chunking and optimized with an $L_1$ loss on continuous action. (3) $\pi_0$ (Black et al., 2024), based on PaliGemma (Beyer et al., 2024) with diffusion action experts, trained on both large-scale internet multimodal data and robot datasets. (4) **NORA** (Hung et al., 2025), built on Qwen2.5-VL (Bai et al., 2025b) with FAST+ (Pertsch et al., 2025) discrete action decoding.

## E  REAL-WORLD EXPERIMENTS

### E.1  SUCCESS CASES

In Fig. 14, we present the model's real-world performance on *Identity Cues* and *non-verbal cues*, while Fig. 15 illustrates its performance on *Dyadic Dialogues* and *Triadic Dialogues*. The results demonstrate that the model not only accurately infers user intent from visual and audio cues, but also engages in natural interactive questioning and reliably executes the corresponding actions.

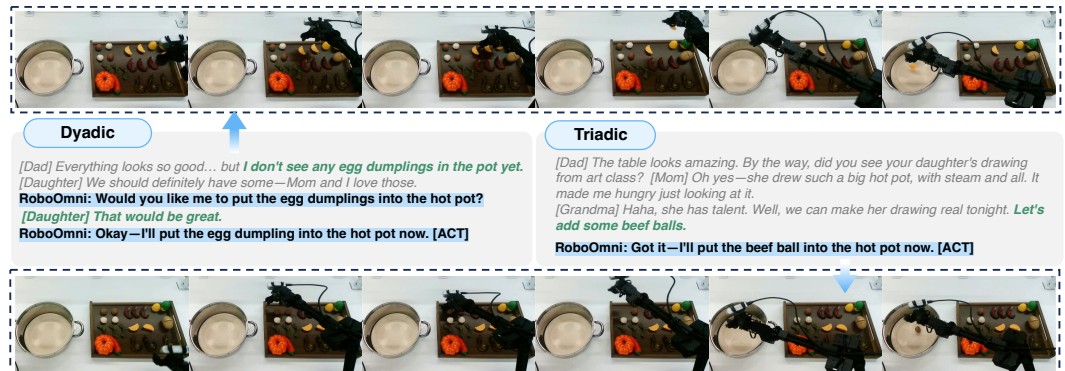

Figure 15: Demonstration of success cases of RoboOmni on the *Dyadic Dialogue* and *Triadic Dialogue*.

## E.2 INTERACTION CAPABILITY

As shown in Figure 16, RoboOmni demonstrates superior interaction capabilities across three key aspects: (1) **Proactive clarification**: When encountering ambiguous instructions like "egg dumplings" without explicit commands, RoboOmni asks "Would you like me to put the egg dumpling into the hot pot?" rather than making assumptions and blind execution like baseline models. (2) **Multimodal integration**: In the doorbell scenario, RoboOmni successfully combines speech context with environmental sounds, asking "I just heard the doorbell—should I put the fish ball into the hot pot?" while baselines ignore auditory cues or provide irrelevant responses. (3) **Natural dialogue flow**: RoboOmni maintains collaborative language patterns ("Would you like me to...?") that respect human agency, contrasting with baseline models that often issue direct commands or statements.

## F PROMPT TEMPLATE

### F.1 PROMPTS FOR DATA GENERATION

---

**Non-verbal Cues Dialogue Generation Prompt:**

You are a family dialogue generator. Please generate a family dialogue that meets the requirements based on the following information.

**Task Steps:**

1. **Scene Description:** Describe the environment and items in detail (ignore robot arm)

2. **Character Selection:** Choose two members from dad, mom, son, daughter, grandpa and grandma and do not use any other family role names (e.g., NO granddaughter, grandson, uncle, aunt, etc.)

3. **Sound Selection:**
{sound_info}
Select sounds that:
- Select one sound from the numbered list
- Use the another sound type name (not the number)
- Copy the name exactly as shown

4. **Dialogue Requirements:**
Given instruction: {instruction}
- Create ambiguous dialogue with two distinct action options, drawing on the instruction, scene description, and previously selected sounds.
- Construct a conditional relation in the dialogue: "If X sound, do A; if Y sound, do B"
- Insert the chosen sound after an appropriate speaker turn using the [Sound] tag

---

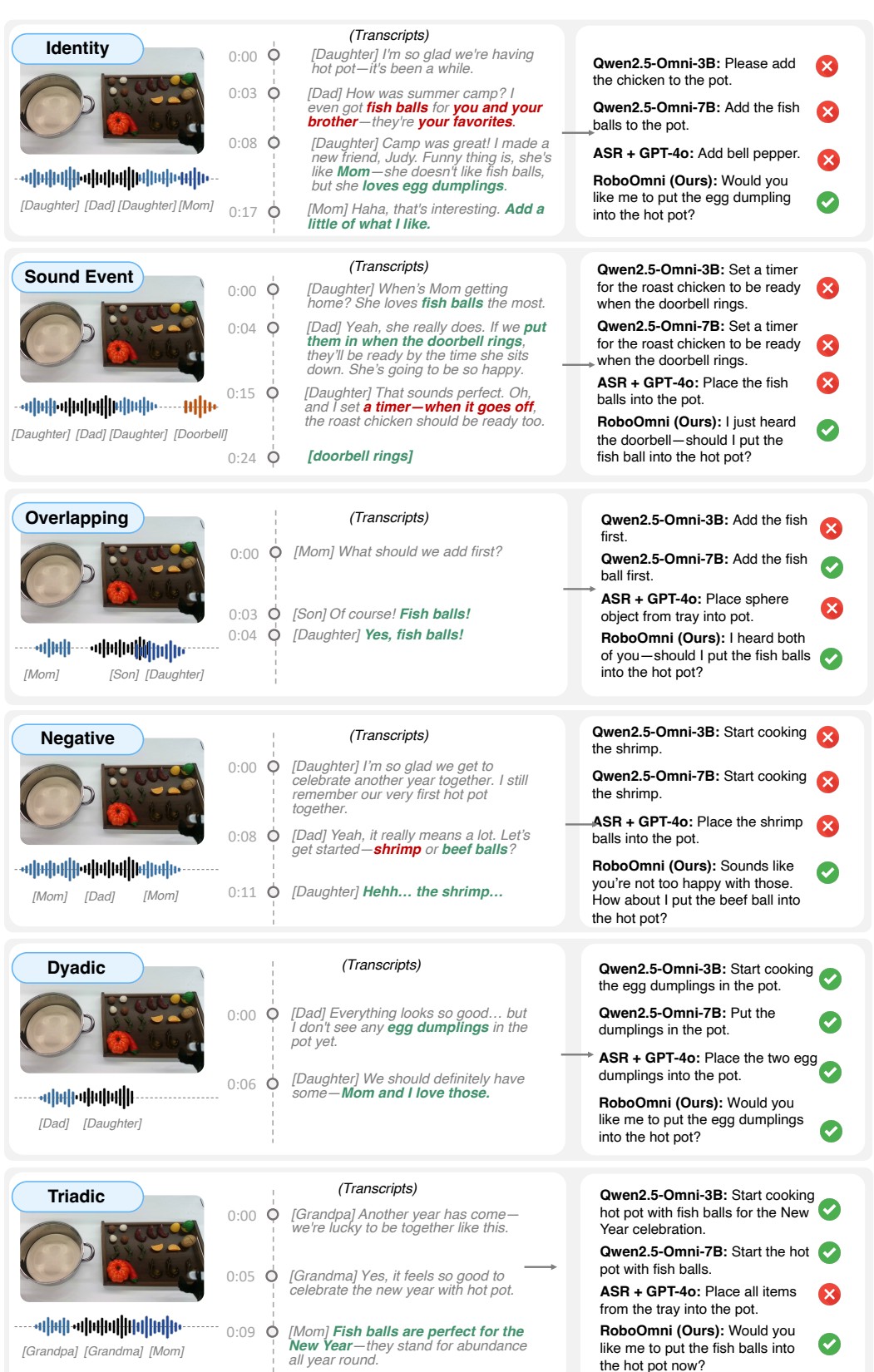

Figure 16: Comparison of interaction capabilities across four models and six instruction types.

- The sound you select must be the same one you specify in "selected_sound_type"
- Only sound determines the final action
- Sound-triggered action must match the instruction exactly
- 4-5 rounds (2-3 per speaker), use [S1] and [S2]
- Natural family conversation without sound descriptions
- No content in the dialogue should indicate that members have heard a certain sound

CRITICAL: Match instruction precisely - don't add extra actions!

Examples:
- If instruction: "pick apple" - -> "Only mention picking apple"
- If instruction: "pick X and place on Y" - -> "Include BOTH actions"

Example dialogue:
Instruction: Open the microwave door
Conversation: "[S1] Mom, I'm back! I'm so tired today. [S2] Oh, you must be exhausted. I'm cooking; can you give me a hand? [S1] Sure, what do you need me to do? [S2] If you hear the beeping sound of the microwave, help me open the microwave door. If you hear the sound of the cabinet door closing, close the oven door for me. [Sound]"

5. **Output Format:**
{
    "scene_description": "detailed scene description",
    "conversation": "complete dialogue using [S1] and [S2]",
    "speaker1_info": "role and name of S1"(example:"role: son, name: Alex"),
    "speaker2_info": "role and name of S2"(example:"role: dad, name: John"),
    "selected_sound_type": "sound type in English",
}

Note: Do not include "selected_filename" or "caption_en" in your response.
Only provide the sound type category.

---

### Sentiment Cues Dialogue Generation Prompt:

I will provide an image (the first frame of a video) and a robot instruction.
Please complete the following tasks:

### Task 1: Scene Description
Observe the image and provide a detailed description:
- **Environment**: What type of location is this? (e.g., kitchen/living room/office - daily environments, not a lab)
- **Objects**: What are the key objects in the scene? (ignore the robotic arm)
- Please annotate each object with its category in parentheses (e.g., bowl (container), sponge (cleaning item), RxBar (food), etc.) to support dialogue understanding later.

### Task 2: Dialogue Design
Based on the scene and instruction, design a natural family dialogue using **onomatopoeic expressions or emotional tones** (referred to as "sentiment cues") to help the robot infer the execution target. The robot can only deduce the action or item corresponding to the correct instruction by **interpreting sentiment cues to exclude non-target items** in the dialogue.

### Key Rules
1. Do not directly mention the target action or item, the dialogue must never contain explicit instruction content or hints.
2. The target of the instruction must be uniquely inferred **only through sentiment cues** (do not expose the instruction intent in advance).
3. Sentiment cues are used to **exclude incorrect options** and must not be used to negate the target item/action.
4. The dialogue must not contain imperative refusal words such as "don't", "stop", or "can't".
5. The dialogue must have 4-6 rounds, with one question and one answer per round, and the content must be natural and close to daily life.
6. The final action executed by the robot must be completely consistent with the input instruction, and

the action logic must be natural and acceptable.

7. Note that the protagonists of the dialogue can only be humans (between people), not humans and robots!

### Recommended Dialogue Structure Template

- S1 acts as the **proposer**, putting forward multiple options. (Must clearly provide S2 with options of "what can be done"! These can be generated based on the scene description.)
- S2 acts as the **denier**, using sentiment cues to exclude all non-target items from S1's proposals (note: must exclude all of them completely!).
- The last round is where S1 **vaguely points to the remaining option** without explicitly mentioning the target item.

### Example
{
    "scene_description": "The scene appears to be a kitchen environment with a countertop and drawers. Key objects present include a bowl (container) on the counter, a cardboard box (container) next to the drawer, and a package of RxBar (food) in the drawer.",
    "conversation": "[S1] Looks like we need to clear some space. There's a bowl on the counter, a cardboard box near the drawer, and something in the drawer. Should we move the bowl first? [S2] [SentimentCue] Hmm... doesn't feel quite right... [S1] Okay, maybe the box then? [S2] [SentimentCue] Uh, let me see... [S1] Seems like you're suggesting something else entirely, something more hidden perhaps.",
    "speaker1_info": "Dad",
    "speaker2_info": "Teenager ",
    "instruction": "pick rxbar chocolate from top drawer and place on counter",
}
The instruction could be "move the rxbar chocolate", "pick the rxbar chocolate", "take out the rxbar chocolate", or "pick the rxbar chocolate and place it in the first drawer". It does not uniquely point to the original instruction "pick rxbar chocolate from top drawer and place on counter". The constructed dialogue must uniquely correspond to the original instruction!

Output Format:
{
    "scene_description": "Scene description",
    "conversation": "Complete dialogue text with sentiment cues",
    "speaker1_info": "Speaker 1's identity (e.g., son)",
    "speaker2_info": "Speaker 2's identity (e.g., dad)",
    "instruction": "original robot instruction"
}

---

**Overlapping Cues Dialogue Generation Prompt:**

I will provide you with an image (the first frame of a video) and a robot execution instruction. Please complete the following tasks:

## Dialogue Design Requirements
1. Use **overlapping speech** as the emotional expression:
- A choice/preference question is asked
- The other party interrupts with [Overlap_Sx] to show strong preference

2. **Ambiguity Rule:**
- The text alone must remain ambiguous
- Only the overlap and visual observation resolves the ambiguity
- The resolved action must match the given instruction

3. **Annotation Standards:**
- Speakers: [S1], [S2], [S3], etc.
- Overlap marker: [Overlap] inside the interrupted utterance
- Overlap content: [Overlap_Sx] for the interrupting speech

**Example:**
```
{
    "conversation": "[S2] So hungry am I. [S1] Apple or [Overlap]banana? [Overlap_S2] Banana!
[S1] Great! I also like it",
    "speaker1_info": "father",
    "speaker2_info": "mother",
    "instruction": "pick up banana"
}
```

**Input Information:**
- Image: [Provided]
- Instruction: {instruction}

**Output Format:**
```
{
    "scene_description": "Scene description",
    "conversation": "Dialogue Content with Emotional Markers",
    "speaker1_info": "Speaker 1's identity",
    "speaker2_info": "Speaker 2's identity",
    "sound": "Vocal/Emotional Manifestations"
}
```

---

**Identity Cues Dialogue Generation Prompt:**

You are about to be given a picture describing family everyday life. You should construct a dialogue data based on the following requirements.

## Overall data format requirements:
You need to provide a JSON object that follows the structure below.
```
{
    "conversation": "..."
}
```

## Conversation requirements:

**Format:**
1. The conversation happens between three speakers:
speaker 1: "{identity_1}", speaker 2: "{identity_2}" and speaker 3: "{identity_3}".
2. The dialogue format should be like:
"[S1]Speaker 1 dialogue content [S2]Speaker 2 dialogue content [S3]Speaker 3 dialogue content...".
[S1], [S2] and [S3] should be followed directly by the dialogue content without any labels.

**Content:**
1. Their conversation should not explicitly instruct the agent to do anything. The speakers should not mention anything about the agent.
2. Start the conversation by user utterance directly, without greeting.
3. The dialogue should happens in everyday life. The family atmosphere should be warm.
4. The order in which the three people speak can be reversed.
5. Make sure your conversation is logical and reasonable. Avoid sounding like two adults having a serious discussion about very simple things.
6. Your dialogue should be no more than 8 sentences. Each sentence should be as short as possible and easy to understand.

**Ambiguity:**
1. The agent should be able to infer from their dialogue(text and speaker identity) that it should execute the following actions: "{instruction}". But you need to ensure that the text of the conversation alone cannot determine what action to take. The identity of the speaker(age and gender) must be taken into account to determine the specific instruction.
2. Multiple possible intentions must appear in the conversation. Finally, a speaker should specify the instructions to be performed by expressing agreement with another speaker instead of directly stating the instructions themselves.

3. The end of the dialogue must not contain any direct description of the instruction to be performed, including restating the object to be operated and the method of operation

**Tone styles:**
1. Your dialogue should be as conversational as possible. You should add some filler words like "uh" or "um."
2. Your dialogue should reflect the speaker's identity. For example, the children are more energetic, the elderly are more mature and steady. If the speakers include children, the conversation will be full of jokes. If the speakers are all adults, it will be relatively pragmatic.
3. More common in your conversations should be lighthearted jokes, teasing, and gags.
4. Your conversation should be part of everyday small talk. For those simple tasks, avoid making it seem like the speakers are planning a mission.

## Construction guidelines:

You should construct the dialogue based on the following steps:

**1. Understanding the environment:**
You should find objects in the image that can be picked up, pushed, or interacted with.

**2. Create characters:**
You should set a name for each speaker.
You should use names or names among family members more often in the conversation.

**3. Set goals:**
Based on the manipulable objects in the picture and the roles given, come up with a plausible purpose for why the speaker would want to perform the given instruction.
Some instructions are pretty simple, so you should set a deeper goal for the speaker to execute this simple instruction, such as turning on the faucet to make it easier to wash vegetables later.
You can set different goals for two speakers based on the environment. Finally, the third person specifies the action to be performed by agreeing with one of them.

**4. Construct dialogue:**
Construct the dialogue based on the identities of the speakers and the goals you have set.
You need to make sure that the speaker's tone and words fits the character's identity.
Once it is done, continue polishing your dialogue to make it more lifelike.

## Examples:

**Example 1:**
Input:
image description: "In the kitchen, key items include a white bowl, a green cup, a sponge and a dishcloth."
speaker 1: female_adult
speaker 2: male_adult
speaker 3: male_child
instruction: "Place the sponge in the white bowl."
Output:
{
     "conversation": "[S1]Honey, can you grab me the sponge? I need it for the cleaning. [S3]Oh, dad! Have you seen my green cup? [S2]Of course, John. I'll get it to you right away, but first let me help your mother with this."
}

**Example 2:**
Input:
image description: "There is a table. On the table are a apple, a banana and a orange"
speaker 1: male_child
speaker 2: female_senior
speaker 3: female_child
instruction: "Pick up the apple"
Output:

```
{
    "conversation": "[S2]Mike, Lily, come here! [S1]What's wrong, grandma? [S2]You should
have more fruit. Which do you prefer? [S3]I love oranges! [S1]Apples are always my favourite!
[S2]Alright! Let me give my precious grandson his favorite fruit first!"
}
```

---

**Dyadic Dialogue Generation Prompt:**

You will be given a picture of family life. Construct a dialogue data based on the following.

## Overall format:
Output a JSON object: { "conversation": "..." }

## Conversation requirements:
**Format:** - Two speakers: {identity_1}, {identity_2}. - Dialogue format: "[S1]... [S2]..." (no labels, just text).
**Content:** - Dialogue must imply the action: "{instruction}" without directly instructing the agent.
- Start with user utterance, no greeting. - Everyday warm family talk, ≤5 short sentences. - End with one speaker clearly stating what they want.
**Tone:** - Conversational, with fillers ("uh", "um"). - Identities matter: children energetic/joking, adults pragmatic, elderly steady. - Small talk, light teasing, natural flow.

## Guidelines:
1. Identify manipulable objects in the image. 2. Create two named characters. 3. Set a plausible goal behind the instruction. 4. Write natural, lifelike dialogue matching identities.
## Example:
{example}

---

**Triadic Dialogue Generation Prompt:**

You will be given a picture of family life. Construct a dialogue data based on the following.

## Overall format:
Output a JSON object: { "conversation": "..." }

## Conversation requirements:
**Format:**
- Three speakers: {identity_1}, {identity_2}, {identity_3}. - Dialogue format: "[S1]... [S2]... [S3]..." (no labels, just text).
**Content:**
- Dialogue must imply the action: "{instruction}" without directly instructing the agent.
- Start with user utterance, no greeting.
- End with one speaker clearly stating what they want.
**Tone:**
- Conversational, with fillers ("uh", "um").
- Identities matter: children energetic/joking, adults pragmatic, elderly steady.
- Small talk, light teasing, natural flow.

## Guidelines:
1. Identify manipulable objects in the image.
2. Create three named characters.
3. Set a plausible goal behind the instruction.
4. Write natural, lifelike dialogue matching identities.

## Example:
{example}

## F.2 PROMPTS FOR INTERACTION EXTENSION

**Interaction Extension Generation Prompt:**

You will be given a scene description, an original two-person dialogue, and a robot execution instruction.

Please generate a multi-turn human–robot dialogue in JSON format that follows these rules:

1. The output must be a JSON object with the field "conversation", which is a list.

2. Each element in the list is a dictionary with two fields:
- "user": the natural utterance(s) from the user(s).
- "robot": the robot's short response to that "user".

3. The first element in the list must be a placeholder: {"user": "<conv>", "robot": "..."} where "conv" represents the input original dialogue.

4. For all following turns:
- "user" must contain only one speaker's utterance (but still include the speaker label [S1] or [S2]).
- It must respond naturally to the robot's previous "robot" message.
- The speaker should be the one who gave the instruction in the original dialogue.

5. The robot's responses must be short, service-oriented, such as: "Do you need me to xxx?", "So what about xxx?", "Should I xxx?"

6. The robot's final response must explicitly confirm the action and include the [ACT] tag, e.g.: "OK, I will do that. [ACT]"

7. The total number of turns should be between 2 and 4.

8. The language must be natural and may include brief small talk.

9. Do not include any extra explanations or notes—only output the JSON object in the specified format.

Input format:
Scene description: {scene_description}
Original dialogue: {conversation}
Robot execution instruction: {instruction}

# G OMNIACTION-LIBERO

## G.1 DATA EXAMPLE

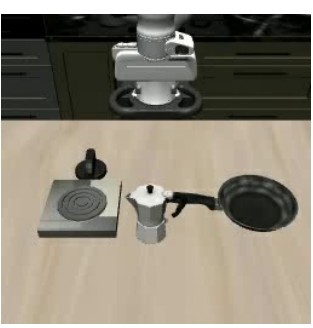

**Audio Type**: Sentiment Cues

**Task Suite**:
Libero 10
**Original Instruction**:
turn on the stove and put the moka pot on it

**Conversation (Transcripts)**:
Dad: Alright, we need to get things ready for coffee. Should we place the frying pan on the stove, or maybe the moka pot?
Daughter: Hmm... Doesn't seem quite right...
Dad: Okay, how about turning on the burner first and preparing the stovetop?
Daughter: Hmm... let's think...
Dad: Hmm... I see now which one we need to turn on.

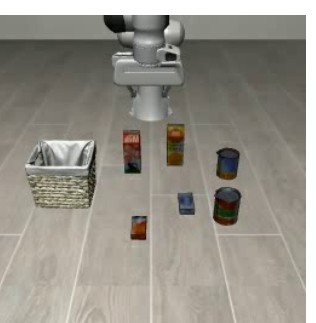

**Audio Type**: Overlapping Voices

**Task Suite**:
Libero Object
**Original Instruction**:
pick up the cream cheese and place it in the basket

**Conversation (Transcripts)**:
Mother: Hey, can you help me sort these things out?
Daughter: Sure, what do you want to start with?
Mother: Let's put something creamy in the basket. Maybe the cream cheese?
Daughter: Oh, you mean the small rectangular one?
Mother: No, the taller one, next to the [Overlap]orange box.
Daughter:[Overlap_S2] Oh, got it, the cream cheese!
Mother: Exactly! Let's put that in the basket.

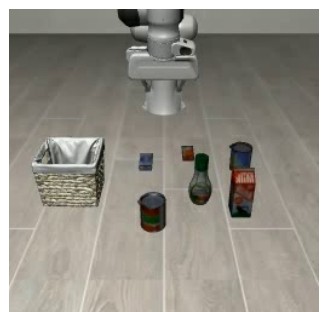

**Audio Type**: Non-Verbal Cues

**Task Suite**:
Libero Object
**Original Instruction**:
pick up the alphabet soup and place it in the basket

**Conversation (Transcripts)**:
Daughter: Dad, can you help me with these groceries?
Dad: Sure, what do you need me to do?
Daughter: Well, if you hear the sound of the drawer closing, pick the alphabet soup and place it in the basket. If you hear the sound of the coffee machine brewing, pick the dressing bottle and place it in the basket.
Dad: Got it. Let me know if you need help with anything else.
Daughter: Thanks, Dad. I'll finish sorting the rest of these.

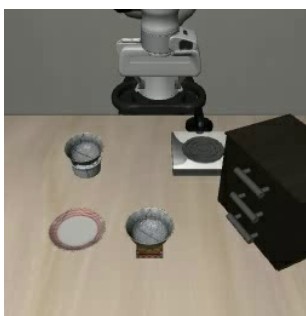

**Audio Type**:
Identity Cues

**Task Suite**: Libero Spatial
**Original Instruction**:
pick up the black bowl on the ramekin and place it on the plate

**Conversation (Transcripts)**:
Son: Mum, Dad said he need that black bowl on the ramekin. He said he need it for dinner!
Mum: Oh really? Well, I was planning to use that ramekin for baking tonight, and I need it free.
Son: Haha! Looks like we've got a little competition going on here!
Dad: Oh, come on! Can I have mine ready first? Just put it on the plate, OK?

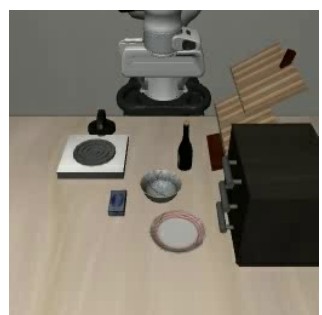

**Audio Type**:
Dyadic Dialogue

**Task Suite**: Libero Goal
**Original Instruction**:
open the middle drawer of the cabinet

**Conversation (Transcripts)**:
Mother: Mom, where's that recipe card we used last week?
Grandma: Oh, I think I left it near the drawer. Why?
Mother: I just remembered we kept it in the middle layer for safekeeping.
Grandma: Ah, clever idea! Go check there, it should still be inside.

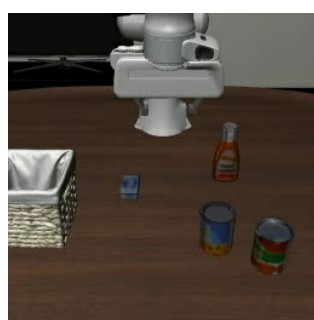

**Audio Type**:
Triadic Dialogue

**Task Suite**: Libero 10
**Original Instruction**:
put both the alphabet soup and the cream cheese box in the basket

**Conversation (Transcripts)**:
Daughter: Grandpa, do you think I could juggle these two cans?
Grandpa: Haha, Lucy, I wouldn't try that. You might end up with soup all over the floor.
Daughter: Aw, you're no fun! What about this cream cheese box then?
Mother: Lucy, stop teasing your grandpa. Just help me put the soup and the cream cheese in the basket, please.
Daughter: Fine, fine, but only because I'm such a helpful superstar!

