# OpenReview forum: "RoboOmni: Proactive Robot Manipulation in Omni-modal Context"
_ICLR.cc/2026/Conference — ICLR 2026 Poster_

### Official Review · Reviewer_2Ge8 · 2025-10-29

**Soundness:** 2
**Presentation:** 3
**Contribution:** 3
**Rating:** 4
**Confidence:** 5

**Summary:**

The paper defines a new setting—cross-modal contextual instructions—where robots infer and confirm user intent from speech prosody, environmental sounds, and vision rather than explicit text, and introduces RoboOmni, a unified Perceiver–Thinker–Talker–Executor model that ingests audio+vision and outputs both dialogue and actions via discrete action tokens (FAST+) in a single autoregressive stream. To supply training data, this paper synthesize OmniAction (≈140k episodes; 5k+ speakers; 2.4k sound events; 640 backgrounds) by GPT-4o–based dialogue rewriting plus multi-speaker TTS with timbre cloning and mixed sound events/backgrounds. They report large gains on a LIBERO-based synthetic-audio benchmark (85.6% vs 25.9% best baseline) and better performance on a small real-speech test with direct audio instructions (76.6% vs 73.8 for π0), along with lower per-inference latency than ASR→VLA cascades; pretraining on OmniAction accelerates SFT.

**Strengths:**

- Timely problem framing. Positioning “contextual instructions” (latent intent from audio+vision, incl. paralinguistics) fills a realistic gap in current VLA assumptions about explicit commands.

- Dataset contribution. OmniAction is large and richly varied across speakers, events, and backgrounds; the pipeline is described with reasonable detail (filtering, GPT-4o rewriting/validation, TTS, overlap mixing, SNR control).

- Strong performance. Big margins on OmniAction-LIBERO-TTS (overall 85.6% vs 25.9 best text/ASR baseline) and competitive performance on real-speech direct commands; end-to-end modeling halves inference latency vs ASR cascades.

**Weaknesses:**

- Synthetic dominates the “new setting.” The core results for contextual instructions are on TTS-synthesized audio and GPT-generated dialogues; the only real-speech evaluation uses direct instructions (not contextual), so the key claim—proactive intent inference from real, messy context—remains under-validated in live conditions.

- Engineering integration over algorithmic novelty. The method mainly composes existing ingredients (Qwen2.5-Omni style encoders, autoregressive LLM “Thinker”, FAST+ action tokens) with a unified likelihood; there’s little new learning principle or analysis specific to proactive intent estimation or uncertainty handling.

- Limited ablations on why it works. We see per-type breakdowns and a pretraining vs from-scratch SFT curve, but no modality ablations (e.g., remove environmental sounds, prosody, or vision), no leave-one-out fusion studies, and no tight control to disentangle dataset scale vs architecture as the main driver.

- External dependence and compute. The pipeline leans on GPT-4o for scripting/validation and large-scale TTS/voice cloning.

- Evaluation scope and safety. Real-robot demos are qualitative and small-scale; there’s no user study on over-eagerness, mis-attribution of speaker intent, or false-positive “proactive” actions, despite identity/overlap being central to the setting.

**Questions:**

- Can you report real-speech, contextual-instruction results (not just direct commands) with humans, including failure/over-intervention rates?

- Which modality drives gains? Please add leave-one-modality-out and prosody/identity ablations to separate benefits of paralinguistics vs vision vs text.

- How sensitive is performance to the FAST+ tokenizer vs continuous action heads, and to chunk length? Any degradation in dexterous or contact-rich tasks?

- What is the wall-clock training and energy cost compared to ASR→VLA cascades tuned for similar accuracy?

- How do you mitigate biases/artifacts from GPT-4o/TTS synthesis (e.g., sentiment exaggeration, timbre leakage) when transferring to real audio? Any domain-gap diagnostics?

---

> ### Author Response · Authors · 2025-11-25
>
> We sincerely thank you for the detailed and insightful comments and appreciate your recognition of our timely problem framing, valuable dataset contribution, and strong empirical performance. We address each concern in detail below.
>
> ### W1 & Q1
> > Synthetic dominates the “new setting.” The core results for contextual instructions are on TTS-synthesized audio and GPT-generated dialogues; the only real-speech evaluation uses direct instructions (not contextual), so the key claim—proactive intent inference from real, messy context—remains under-validated in live conditions.
>
> > Can you report real-speech, contextual-instruction results (not just direct commands) with humans, including failure/over-intervention rates?
>
>
> Thank you for highlighting the importance of validating contextual-instruction understanding under real human speech.
> - Importantly, our **Intent Recognition Capability** experiment (Sec. 5.5) already uses **real-human recorded speech**, not TTS. It contains 180 human-spoken contextual dialogues (6 categories × 30 samples). RoboOmni achieves 88.9% accuracy, outperforming ASR+GPT-4o and Qwen-Omni under identical conditions. This confirms that our intent-understanding ability is not limited to synthetic data.
> - Beyond intent recognition, we have added **a new quantitative real-speech contextual-instruction evaluation** in the revision (Fig. 5), consisting of 18 tasks across 6 contextual-instruction categories, each executed for 10 trials with average task success reported. We recruited **10 human volunteers** to design contextual dialogues (not GPT-generated) and record them. These recordings naturally include hesitation, prosody changes, background noise, and indirect/implicit requests — closely matching the “real, messy context”.
> - Results show that RoboOmni maintains strong performance under real human contextual instructions: Task success, **achieving 73.9% task success, substantially outperforming the best ASR+VLA baseline (52.2%)**. These results are consistent with our synthetic-audio benchmark, indicating that the model’s ability to infer intent does transfer to in-the-wild speech.
> -  We have updated this experiment in the new version.
>
> | Model        | Sentiment | Non-verbal | Identity | Overlapping | Dyadic | Triadic | Avg  |
> |--------------|-----------|------------|----------|-------------|--------|---------|------|
> | ASR+OpenVLA  | 23.3      | 30.0       | 23.3     | 40.0        | 50.0   | 36.7    | 33.9 |
> | ASR+NORA     | 16.7      | 26.7       | 33.3     | 43.3        | 60.0   | 43.3    | 40.6 |
> | ASR+Pi0      | 46.7      | 36.7       | 40.0     | 53.3        | 70.0   | 66.7    | 52.2 |
> | Ours         | **76.7**      | **66.7**       | **73.3**     | **76.7**        | **80.0**   | **70.0**    | **73.9** |
>
>
> ### W2
> > Engineering integration over algorithmic novelty. The method mainly composes existing ingredients (Qwen2.5-Omni style encoders, autoregressive LLM “Thinker”, FAST+ action tokens) with a unified likelihood; there’s little new learning principle or analysis specific to proactive intent estimation or uncertainty handling.
>
> Our contributions go far beyond “engineering integration”; they lie in a new problem formulation and a new unified probabilistic modeling approach.
> - **New problem formulation:** Prior VLA systems all assume explicit instructions. We first introduce **cross-modal contextual instructions**, requiring robots to infer latent user intent from prosody, identity cues, overlapping dialogue, non-verbal sound events, and vision. This setting has not been studied before and fundamentally changes the task definition.
> - **New unified modeling principle:** To support this new setting, we do not simply concatenate multiple modalities. Instead, we place visual hiddens, audio hiddens, text tokens, and action tokens into a single autoregressive token space, allowing them to participate in the same reasoning chain. This avoids the inherent information loss in ASR → text → VLA cascades, which inevitably discard prosody and sentiment cues, break temporal alignment, and create semantic discontinuities.
> - Large-scale cross-modal dataset enabling alignment. We further construct a large-scale vision–audio–action dataset using a comprehensive synthesis pipeline, enabling effective cross-modal alignment through data-driven learning. The resulting **140k real-world cross-modally grounded trajectories** represent a substantial contribution to the field and are essential for learning proactive intent inference.

---

> ### Author Response · Authors · 2025-11-25
>
> ### W3 & Q2
> > Limited ablations on why it works. We see per-type breakdowns and a pretraining vs from-scratch SFT curve, but no modality ablations (e.g., remove environmental sounds, prosody, or vision), no leave-one-out fusion studies, and no tight control to disentangle dataset scale vs architecture as the main driver.
>
> > Which modality drives gains? Please add leave-one-modality-out and prosody/identity ablations to separate benefits of paralinguistics vs vision vs text.
>
> - To isolate the contribution of each modality, we perform input-controlled ablations by dropping the input of specific modalities. Concretely, we evaluate intent recognition under four settings: (1) **No vision** (visual input withheld, model receives only audio) (2) **No audio** (audio input dropped, model receives only vision) (3) **No paralinguistics** (all audio re-recorded by a single neutral speaker without prosody, emotion, or non-verbal sound events) (4) **Full Input** (ours, vision + audio with prosody)
> - From the results in Table 3, we observe that (1) **Audio carries the largest marginal effect**, since removing it removes the actual instruction. (2) **Vision is essential** whenever contextual instructions reference spatial relations or visual attribution—e.g., understanding requests such as “*I want the item to the left of object A*” or “*pass me the red one*”. (3) **Paralinguistics provide important disambiguation cues.** For example, inferring intent from the speaker’s emotional tone, using vocal characteristics to identify the user and adapt to their preferences, or interpreting environmental sound events (e.g., an oven timer ringing) as triggers for the appropriate action.
> - The ablations confirm that no single modality alone explains RoboOmni’s gains. Performance degrades substantially when any modality is removed, demonstrating that contextual-instruction understanding is intrinsically multi-modal.
> - We have updated this experiment in the new version.
>
> | **Setting**              | **Accuracy (%)** |
> |--------------------------|------------------|
> | **Full Input (ours)**    | **88.89**        |
> | w/o vision               | 58.89            |
> | w/o audio                | 11.11            |
> | w/o paralinguistics      | 50.56            |
>
> ### W4
> > External dependence and compute. The pipeline leans on GPT-4o for scripting/validation and large-scale TTS/voice cloning.
>
> - Thank you for the suggestion. Using external tools such as GPT-4o and TTS/voice cloning to assist data generation is a common and practical approach to reduce time and human cost, and the value of such data ultimately depends on real-world performance rather than its source.
> - To ensure this, we evaluate with **real human speech** across (1) the direct-instruction experiments in Sec. 5.3, (2) the intention-recognition experiments in Sec. 5.5, and (3) the additional real-world contextual-instruction experiment added in the rebuttal W1. The results consistently show that: (1) On intent recognition, our model achieves 88.89% accuracy, clearly outperforming ASR+GPT-4o and Qwen-Omni under identical input conditions. (2) On real-robot execution, our model reaches 73.9% task success, substantially surpassing the best ASR+VLA baseline (52.2%). These findings demonstrate that the tool-assisted data is highly effective for training reliable, real-world robotic behavior. What's more, **our method performs strongly even under real human speech and real-world conditions without any external dependence on GPT/TTS-generated data.**

---

> ### Author Response · Authors · 2025-11-25
>
> ### W5
> > Evaluation scope and safety. Real-robot demos are qualitative and small-scale; there’s no user study on over-eagerness, mis-attribution of speaker intent, or false-positive “proactive” actions, despite identity/overlap being central to the setting.
>
> - Thank you for the helpful suggestions. In addition to the qualitative demonstrations, we have added **a new quantitative real-robot evaluation** (reported in the response to W1/Q1 and now included in Sec. 5.5, Fig. 5), covering 18 tasks × 10 trials with human-spoken contextual instructions. RoboOmni achieves **73.9%** task success, substantially outperforming the best ASR→VLA baseline (52.2%), providing quantitative evidence of its real-world effectiveness under identity cues, overlapping speech, and non-verbal context.
> - We further include **a detailed error-type breakdown** (Fig.11 in Sec. 5.6), separating intention-level errors (42.6%) from action-level errors (57.4%). Within intention-level failures, we quantify identity-related and overlapping-speech misclassification (14.9% and 2.1%), showing that such cases do occur but remain limited. Importantly, the majority of failures arise from low-level manipulation issues (e.g., grasp failure 29.8%, pose drift 14.9%) rather than unsafe proactive behaviors.
> - We have updated it in the new version to make it clearer.
>
> | Cat. | Type                |  |
> |------|---------------------|-----|
> | **Intent (42.6%)** | ID Attribution       | 14.9% |
> |      | Overlap Speech      | 2.1%  |
> |      | Non-verbal Cue      | 6.4%  |
> |      | Sentiment Cue       | 12.8% |
> |      | Dialog Flow         | 6.4%  |
> | **Action (57.4%)** | Grasp Failure        | 29.8% |
> |      | Pose Drift          | 14.9% |
> |      | Reachability        | 4.3%  |
> |      | Others              | 8.5%  |
>
>
> ### Q3
> > How sensitive is performance to the FAST+ tokenizer vs continuous action heads, and to chunk length? Any degradation in dexterous or contact-rich tasks?
>
> - Our model operates as a single unified autoregressive policy that generates both text tokens and action tokens in the same token stream. Under this formulation, a tokenized action representation (e.g., FAST+) is the most natural way to maintain architectural consistency, as it lets the policy interleave reasoning, dialogue, and low-level actions within the same decoding stream. Introducing a separate continuous action head would create a hybrid architecture and change the problem setting rather than isolating a single factor for ablation.
> - Regarding execution quality, prior work on FAST+ [1] has shown that discrete and continuous parameterizations achieve **comparable control fidelity**, including on contact-rich or dexterous tasks. Our work builds on these findings, and in our domains we did not observe degradation attributable to the tokenized action representation.
>
> [1] FAST: Efficient Action Tokenization for Vision-Language-Action Models
>
> ### Q4
> > What is the wall-clock training and energy cost compared to ASR→VLA cascades tuned for similar accuracy?
>
> - As shown in our new **leave-one-modality-out ablations (Q2 response)**, removing paralinguistic cues (prosody, speaker identity, environmental sounds) leads to a substantial drop in intent recognition accuracy. An ASR→VLA cascade—by construction—does **not** preserve these paralinguistic signals, since standard ASR outputs text only. Therefore, an ASR→VLA pipeline cannot realistically be “tuned for similar accuracy” in contextual-instruction tasks, because the missing paralinguistic information is precisely what enables intent inference in our setting.
> - In terms of the efficiency, As reported in Fig. 8, our unified autoregressive model achieves **significantly lower inference** latency than an ASR→VLA cascade with a comparable backbone size. The improvement largely comes from avoiding a separate ASR pass and eliminating the need to synchronize two modules during rollouts. This reflects the practical advantage of end-to-end modeling in interactive robotic settings.

---

> > ### Author Response · Authors · 2025-11-25
> >
> > ### Q5
> > > How do you mitigate biases/artifacts from GPT-4o/TTS synthesis (e.g., sentiment exaggeration, timbre leakage) when transferring to real audio? Any domain-gap diagnostics?
> >
> > - We applied multiple steps to reduce potential artifacts and improve diversity/realism: (1) **Prompt diversification**. Varied templates encourage different conversational styles and reduce script homogeneity. (2) **Temperature elevation**. We use a higher sampling temperature to reduce overly rigid or repetitive text generation. (3) **Redundancy filtering using ROUGE-L**. Dialogues with high similarity to existing ones are removed, increasing dataset heterogeneity.(4) **Human verification of dialogue realism**. A subset of dialogue text is manually checked to confirm naturalness and correctness.
> > - We also introduced an additional human evaluation for the TTS audio: We sampled 200 TTS clips and asked three annotators to judge whether the intended attributes were correctly expressed (e.g., child voice, emotional tone, speaker style). Human accuracy was **98.7%**, indicating that the TTS outputs match the specified attributes reliably.
> > - Moreover, our **real-human speech evaluations** serve as an explicit domain-gap diagnostic.  The large-scale synthetic data is primarily used to achieve **robust cross-modal alignment between vision and audio**, enabling the model to learn diverse contextual-instruction patterns. We then verify this capability on real human speech, including (1) direct human-spoken instructions (Tab. 2) and (2) the real-speech contextual-instruction study added in the rebuttal W1 (Fig. 5 in the revision).  In both cases, RoboOmni maintains competitive performance, indicating that learning does not overfit to TTS-specific patterns.

---

> > > ### Author Response · Authors · 2025-11-27
> > >
> > > Dear Reviewer 2Ge8,
> > >
> > > Thank you again for your thoughtful and constructive review. We would like to briefly note that several additional experiments and clarifications—particularly the new real-speech contextual-instruction evaluation, modality-drop ablations, and error-type analysis—have been incorporated in our rebuttal to directly address the concerns you raised.
> > >
> > > If any part of our response would benefit from further clarification, we are very happy to provide more details. We sincerely appreciate your time and engagement during the discussion period.
> > >
> > > Best regards,
> > >
> > > Authors of #7282

---

### Official Review · Reviewer_xDAu · 2025-10-30

**Soundness:** 3
**Presentation:** 3
**Contribution:** 3
**Rating:** 6
**Confidence:** 4

**Summary:**

The paper targets proactive intention estimation, action planning, and execution for embodied agents in realistic human-robot interaction. It introduces a large-scale synthetic multimodal dataset built on Open-X with off-the-shelf text and audio models, and proposes RoboOmni, a unified framework that combines omni-modal perception with low-level action generation. Experiments suggest strong performance on proactive intention recognition and end-to-end planning/execution.

**Strengths:**

- The scenarios are well-chosen, practically relevant, and aligned with next-generation robots operating in realistic human-robot interaction settings.
- The constructed dataset is likely to be valuable for the community, providing broad multimodal supervision that can stimulate research on proactive robotic behavior.

**Weaknesses:**

- Neither the dataset’s realism/relevance nor the model’s performance appears to be evaluated by humans. This limits claims about practical usability and interaction quality. While Section 5.3 covers direct human audio instructions, this differs from the complex multimodal contexts central to the paper’s claims.
- The dataset may inherit biases from off-the-shelf models used during curation. This paper does not analyze or mitigate such biases.
- The setup, tuning, and fairness of baseline comparisons are insufficiently described, making it hard to interpret performance gaps. Please see the Question section for details.

**Questions:**

- Please elaborate on the procedure for “removing trivial samples with low-information visual states.” How do you quantify or detect low information? What thresholds and ablation evidence support the chosen criteria?

- Does the model feed its own intermediate outputs back into the pipeline? If so, are those outputs fed back as text or as audio?

- Were baselines in Table 1 tuned on RoboAction or a simplified variant (e.g., audio converted to text)? If not, low performance may reflect distributional mismatch rather than model limitations. Similarly, were the planner and VLA components of the baselines in Figure 8 tuned for these tasks?

---

> ### Author Response · Authors · 2025-11-25
>
> Thank you for the constructive comments and for highlighting that our proposed problem setting aligns with next-generation human–robot interaction, as well as the value of our multimodal dataset. We respond to the remaining questions and concerns in detail below.
>
> ### W1
> > Neither the dataset’s realism/relevance nor the model’s performance appears to be evaluated by humans. This limits claims about practical usability and interaction quality. While Section 5.3 covers direct human audio instructions, this differs from the complex multimodal contexts central to the paper’s claims.
>
> - We clarify that we do conduct human evaluation on both dataset realism and model performance. (1) **For the dataset, we performed human checks** on sampled OmniAction dialogues and TTS outputs to verify intent correctness and phenomenon fidelity. Human evaluators achieved 98.7% agreement on the validation samples (detailed in Section 3.2 and Appendix C.4). (2) For model evaluation, the **intent-recognition experiments** in Sec. 5.5 are human-assessed.
>
> - Additionally, we include **a new quantitative real-world contextual-instruction experiment** (Fig 5 in our revised version), where 10 volunteers created and recorded their own contextual dialogues and audio; the interactions are fully unscripted and **human-evaluated**.  Results show that RoboOmni sustains strong performance under real human contextual instructions, achieving 73.9% real-robot task success—substantially outperforming the best ASR+VLA baseline (52.2%). These findings are consistent with our synthetic-audio benchmark, indicating that the model’s intent understanding and action execution capabilities remain effective under  in-the-wild human speech. We have updated this experiment in the new version.
>
> | Model        | Sentiment | Non-verbal | Identity | Overlapping | Dyadic | Triadic | Avg  |
> |--------------|-----------|------------|----------|-------------|--------|---------|------|
> | ASR+OpenVLA  | 23.3      | 30.0       | 23.3     | 40.0        | 50.0   | 36.7    | 33.9 |
> | ASR+NORA     | 16.7      | 26.7       | 33.3     | 43.3        | 60.0   | 43.3    | 40.6 |
> | ASR+Pi0      | 46.7      | 36.7       | 40.0     | 53.3        | 70.0   | 66.7    | 52.2 |
> | Ours         | **76.7**      | **66.7**       | **73.3**     | **76.7**        | **80.0**   | **70.0**    | **73.9** |
>
>
> ### W2
> > The dataset may inherit biases from off-the-shelf models used during curation. This paper does not analyze or mitigate such biases.
>
> - Thank you for raising this point. In practice, we found that a combination of automated validation and human verification was effective in keeping such issues minimal. As described in Sec. 3.2, all synthesized dialogues undergo an automated GPT-based consistency check that filters out samples with unnatural phrasing, incoherent speaker roles, or incorrect content.
> - Beyond automated checks, we also performed human reviews on a curated subset of dialogues and TTS outputs to confirm naturalness, appropriateness, and alignment with everyday conversational patterns. In addition, the dataset was intentionally constructed with diverse prompt templates, varied domains, and heterogeneous in-context examples, reducing the likelihood that the data would inherit narrow stylistic patterns from any single model.
> - Finally, our real-speech evaluations—both the direct-instruction tests (Sec. 5.3) and the fully unscripted contextual-instruction experiments with 10 volunteers (Sec. 5.4 in the revision) —demonstrate strong performance under human-produced audio. This suggests that residual artifacts in the synthetic data do not materially affect real-world behavior or interaction quality.
>
>
> ### W3
> > The setup, tuning, and fairness of baseline comparisons are insufficiently described, making it hard to interpret performance gaps. Please see the Question section for details.
>
> Thank you for the constructive suggestions regarding the presentation of the paper. We address all questions (**Q1, Q2, Q3**) in the point-by-point responses below.

---

> ### Author Response · Authors · 2025-11-25
>
> ### Q1
> > Please elaborate on the procedure for “removing trivial samples with low-information visual states.” How do you quantify or detect low information? What thresholds and ablation evidence support the chosen criteria?
>
> - The “low-information visual states” filtering step is applied to avoid trivial scenarios where visual context provides almost no disambiguation signal—such as scenes with a single object, limited actionable possibilities, or backgrounds that make contextual intent inference meaningless.
> - To operationalize this, we use a GPT-based evaluator that produces a qualitative assessment of visual informativeness by examining object count, visibility of potential interaction targets, and the presence of cues needed for ambiguous or indirect instructions. Rather than imposing a fixed numeric threshold, we adopt a binary judgment (“suitable / not suitable for contextual-instruction synthesis”) to simplify the decision process and avoid threshold tuning.
> - Human reviewers also inspected paired filtered/non-filtered samples: they consistently preferred dialogues generated on filtered scenes because these scenes naturally support richer ambiguous or indirect intent, which is central to the contextual-instruction setting.
>
> ### Q2
> > Does the model feed its own intermediate outputs back into the pipeline? If so, are those outputs fed back as text or as audio?
> - Yes. Since RoboOmni is a multi-turn interaction model, its intermediate outputs are naturally fed back into the next round.
> - Our current implementation operates in a half-duplex setting, which is the most mainstream and controllable design approach for current speech language models. Half-duplex interaction provides greater stability, safety, and resource efficiency by avoiding model self-interference, reasoning confusion, and temporal management complexity. However, this design means our model can only listen or speak at given time, rather than supporting simultaneous listening and speaking (the advantage of full-duplex systems). To minimize user waiting time without requiring the model to complete its full speech output before proceeding to the next reasoning step, we directly use the text generated by the thinker module as input to RoboOmni for subsequent processing.
> - Enabling full-duplex interaction is a natural direction for future work. This can be done by adding a lightweight, plug-and-play VAD (voice activity detection) module, a standard component in speech-interactive systems (e.g., Qwen3-Omni), which would allow the robot to detect user speech and respond concurrently.
>
> ### Q3
> > Were baselines in Table 1 tuned on RoboAction or a simplified variant (e.g., audio converted to text)? If not, low performance may reflect distributional mismatch rather than model limitations. Similarly, were the planner and VLA components of the baselines in Figure 8 tuned for these tasks?
>
> - The models in Table 1 are publicly released direct-instruction VLAs evaluated without additional tuning on OmniAction, consistent with standard practice in prior VLA benchmarks. Table 1 is intended to illustrate that existing VLA can handle only explicit direct instructions and are fundamentally incapable of operating in the contextual-instruction setting introduced in this work, a conclusion supported by the observed results.
> - For a fair comparison under aligned instruction distributions, Table 2 reports results on direct instructions. All methods—including ASR→VLA pipelines—receive identical text inputs. Even under this fully matched setup, baseline VLAs remain highly sensitive to minor linguistic variations introduced during ASR transcription, a known issue in prior VLA work. This highlights a structural limitation of cascaded ASR→VLA systems.
> - To further probe whether a stronger external module could compensate for these limitations in Table 1, Figure 8 examines a planner–controller pipeline where Qwen2.5-Omni consumes raw audio and text-based VLAs act as controllers. We intentionally avoid additional fine-tuning because the goal is to test whether high-level planning alone can bridge the modality gap. The results show that even when Qwen2.5-Omni consumes raw audio and existing VLAs act only as controllers, the planner–controller separation prevents contextual cues from reaching the action module.
> - In addition, our real-robot results (Sec. 5.5 / W1 response) use **SFT-tuned baselines on the same conversational instruction data** as RoboOmni to ensure distributional consistency.  Even under this matched training setup, RoboOmni still achieves a substantial advantage. This is because RoboOmni directly reasons over raw audio + vision, preserving prosody, identity, and non-verbal events essential for contextual-intent inference—signals that ASR-based pipelines inherently discard.

---

> > ### Author Response · Authors · 2025-11-27
> >
> > Dear Reviewer xDAu,
> >
> > We appreciate the thoughtful comments you shared. In response, we have added the real-speech contextual-instruction evaluation and provided clearer explanations of dataset filtering, potential artifacts, and baseline setup. Your suggestions helped us present the work more transparently. We hope the added revisions address the points you raised, and we remain glad to clarify further details whenever needed.
> >
> > Best regards,
> >  Authors of #7282

---

### Official Review · Reviewer_GVRu · 2025-11-01

**Soundness:** 3
**Presentation:** 3
**Contribution:** 4
**Rating:** 8
**Confidence:** 4

**Summary:**

The manuscript proposes a multimodal fusion framework that integrates text, speech/dialogue, vision, and environmental sounds to support robotic reasoning and decision-making. Compared with using text as the sole communication bridge, the unified end-to-end approach achieves stronger performance with lower control latency, supported by both simulated and real-world experiments.

**Strengths:**

- Clear writing and well-structured presentation.

- Promised release of a robotic dataset, extensive experiments across diverse tasks, and large-scale training.

- An “omni” framework for cross-modal contextual instructions that explicitly includes environmental event/background sounds; comparisons to ASR-based pipelines highlight the benefits of an end-to-end model that can handle overlapping speech.

- Strong empirical results, with ≈60% average success rates exceeding other baselines.

**Weaknesses:**

- Limited discussion of the method and training for fusing multimodal sensory inputs.

- Ablations on modality contributions are missing: it remains unclear how much each cue (prosody/identity, non-verbal audio, vision) contributes. Please add drop-modality ablations (e.g., audio-w/o-prosody, no non-verbal, no vision) and alignment-window studies; current results separate instruction types but not modalities within the architecture.

- Task descriptions are brief; it is hard to understand the challenges without consulting references.

- Baseline fairness: many baselines rely on ASR (Whisper) or ground-truth text, which can be disadvantaged when paralinguistic information matters. One can retain a language-based architecture but add a separate “translation module” to detect and convert such information [1] into prompt context (e.g., simple classifiers for emotion or dialogue overlap, then embed tokens like “[overlapped_dialogue] [disappointed]”). This increases complexity but is a straightforward way to extend text-based models to richer context.

- Minor: the related-work section could be expanded for a more comprehensive background; prior work [2, 3] also explores multimodal decision-making including environmental sound.

References

[1] Yamakawa, N., Takahashi, T., Kitahara, T., Ogata, T., & Okuno, H. G. (2011, June). Environmental sound recognition for robot audition using matching-pursuit. In International Conference on Industrial, Engineering and Other Applications of Applied Intelligent Systems (pp. 1-10). Berlin, Heidelberg: Springer Berlin Heidelberg.

[2] Zhao, X., Li, M., Weber, C., Hafez, M. B., & Wermter, S. (2023, October). Chat with the environment: Interactive multimodal perception using large language models. In 2023 IEEE/RSJ International Conference on Intelligent Robots and Systems (IROS) (pp. 3590-3596). IEEE.

[3] Liu, Z., Chi, C., Cousineau, E., Kuppuswamy, N., Burchfiel, B., & Song, S. (2024). Maniwav: Learning robot manipulation from in-the-wild audio-visual data. arXiv preprint arXiv:2406.19464.

**Questions:**

- Following W1 and W2: environmental event sounds can be sparse compared to speech or vision—will the model ignore this feature during fusion? How is this addressed and evaluated? Also, how is synchronization among modalities handled? What is the performance change if environmental sounds are removed?

- The experiments show large gains over baselines. Could the authors provide insights into why those baselines fail and why RoboOmni succeeds?

---

> ### Author Response · Authors · 2025-11-25
>
> We are grateful for your constructive and encouraging comments and for highlighting our clear presentation, large-scale data contribution, unified end-to-end omni-model, and strong experimental results. We provide detailed responses to each concern below.
>
> ### W1
> > Limited discussion of the method and training for fusing multimodal sensory inputs.
>
> Thank you for suggestions on paper writing. We have expanded the description of our multimodal fusion mechanism (see Section 4.1 & 4.2 in the revised version). Each modality is first encoded by its own encoder (text tokenizer, vision encoder, audio encoder) and then projected into a shared embedding space, forming a unified token sequence. This sequence is processed by a single Transformer, where full cross-modal self-attention enables fine-grained fusion among visual cues, prosody, overlap patterns, and environmental audio events. All modalities are optimized jointly under the same behavior-cloning objective, ensuring they are fused end-to-end rather than used in a cascaded manner.
>
> ### W2
> > Ablations on modality contributions are missing: it remains unclear how much each cue (prosody/identity, non-verbal audio, vision) contributes. Please add drop-modality ablations (e.g., audio-w/o-prosody, no non-verbal, no vision) and alignment-window studies; current results separate instruction types but not modalities within the architecture.
>
> - Thanks for your advice. We have added new modality-drop ablations as requested. Concretely, we evaluate intent recognition under four settings: (1) *No vision*: visual frames removed, model receives only audio; (2) *No audio*: all audio removed, model receives only vision; (3) *No paralinguistics*: the same speech content is preserved but re-recorded by a single neutral speaker without prosody, identity cues, or non-verbal sound events; (4) *Full input (ours)*: vision + full audio including paralinguistics.
> - The Table 3 in the revision now reports: (1) Removing **audio** severely degrades instruction understanding, particularly for overlapping-speech and non-verbal categories. (2) Removing **vision** also causes significant degradation on spatial-reasoning tasks. (3) Removing **paralinguistics** specifically harms identity-sensitive and emotional-intent instructions.
> - We thank the reviewer for pointing this out—this ablation makes the contribution substantially clearer, showing that each modality is essential and that the model indeed leverages them rather than ignoring sparse cues. We have updated this experiment in the new version.
>
> | **Setting**              | **Accuracy (%)** |
> |--------------------------|------------------|
> | **Full Input (ours)**    | **88.89**        |
> | w/o vision               | 58.89            |
> | w/o audio                | 11.11            |
> | w/o paralinguistics      | 50.56            |
>
>
> ### Q1
> > Following W1 and W2: environmental event sounds can be sparse compared to speech or vision—will the model ignore this feature during fusion? How is this addressed and evaluated? Also, how is synchronization among modalities handled? What is the performance change if environmental sounds are removed?
>
>
> - In our task taxonomy, the **non-verbal cues** category is explicitly designed to capture cases where environmental event sounds are the **primary carriers of intent** (e.g., alarms, boiling water, appliance beeps). These sounds are not isolated background noise—they are **semantically tied to task intent**, so the model cannot simply ignore them. They appear sparsely but decisively, and models that ignore them fail to infer the correct action. For example:
>   -  (1) the user instructs the robot to “*add the meatballs when the kitchen timer beeps,*” making the timer sound a mandatory trigger;
>   -  (2) the sound of *boiling water reaching a rapid boil* indicates that hotpot preparation is ready, prompting the robot to ask whether to add the dumplings from the serving plate.
>
>  These cues are typical in real homes and serve as implicit triggers that cannot be recovered from text or vision alone.
>
> - To verify that the model truly relies on these sparse but critical events, we conduct an ablation on the non-verbal category by removing only environmental event sounds used in these tasks. Under this setting, the intention recognition performance drops drastically (from 90% → 23.3%). This targeted ablation demonstrates that the model indeed makes use of sparse acoustic events rather than ignoring them.

---

> > ### Author Response · Authors · 2025-11-25
> >
> > ### W3
> > > Task descriptions are brief; it is hard to understand the challenges without consulting references.
> >
> > Thank you for pointing this out. We agree that clearer task descriptions improve readability without requiring external references. In the revised version, we have added a new Section 4.1 to explicitly define the contextual-instruction task and highlight its core challenges. In addition, Section 3.1 has been included with concise explanations of each task category, together with their inherent difficulties (e.g., intent ambiguity, multi-party dialogue overlap, non-verbal audio triggers).
> >
> > ### W4
> > > Baseline fairness: many baselines rely on ASR (Whisper) or ground-truth text, which can be disadvantaged when paralinguistic information matters. One can retain a language-based architecture but add a separate “translation module” to detect and convert such information [1] into prompt context (e.g., simple classifiers for emotion or dialogue overlap, then embed tokens like “[overlapped_dialogue] [disappointed]”). This increases complexity but is a straightforward way to extend text-based models to richer context.
> >
> > We clarify how our baselines already approximate  the suggested “translation module” and why they still fall short.
> > - **Text baseline.** For overlapping-dialogue tasks, our ground-truth transcripts already contain explicit overlap markers (e.g., `[S1] ... [Overlap] ... [Overlap_S2] ...`). Even with these symbolic cues, the text-only baseline still performs poorly on overlap instructions, indicating that injecting tokens alone is insufficient.
> >
> > - **Qwen-Omni-based baseline.** As further analyzed in Figure 8, the Qwen-Omni → VLA cascade functions similarly to the proposed separate “translation module”: it captures prosody, emotion, and overlap before passing enriched text to the policy. Despite its strong speech understanding capability, its embodied intent recognition accuracy is **only ~50%**, and performance on environmental-sound triggers is even lower.
> >
> > These results show that even with symbolic overlap cues or a powerful paralinguistic translation front-end, cascaded text pipelines remain significantly weaker than our unified multimodal fusion approach.
> >
> >
> > ### W5
> > > Minor: the related-work section could be expanded for a more comprehensive background; prior work [2, 3] also explores multimodal decision-making including environmental sound.
> >
> > Thank you for pointing this out. We have incorporated the suggested references and expanded the related-work section (Line 120-122) accordingly.
> >
> > ### Q2
> > > The experiments show large gains over baselines. Could the authors provide insights into why those baselines fail and why RoboOmni succeeds?
> >
> > - The baselines fail mainly because they rely on **cascaded text pipelines** (text or ASR → VLA), which compress rich multimodal cues—prosody, overlapping speech, and environmental events, and vision observation—into a limited symbolic representation. This causes two fundamental issues: (1) **Loss of fine-grained signals** (e.g., emotional tone, overlap timing, subtle environmental cues), and (2) **Error propagation** from ASR or symbolic translation to downstream control.
> >
> > - RoboOmni succeeds because it performs **unified, end-to-end fusion** of vision, speech, non-verbal audio, and action in a single Transformer. All modalities are aligned in the hidden space and jointly optimized under the same supervision, enabling the model to attend to continuous prosodic cues, environmental triggers, and visual context without information bottlenecks. This eliminates cascading errors and preserves multimodal temporal structure, leading to substantially higher intent-understanding and robotic success rates.

---

> > > ### Author Response · Authors · 2025-11-27
> > >
> > > Dear Reviewer GVRu,
> > >
> > > Thank you for the time and constructive insights you provided. We have updated the revision to reflect your feedback, including the added drop-modality ablations and the expanded descriptions of the fusion mechanism, experimental setup, and task definition. We greatly appreciate your comments, which helped us improve the clarity and completeness of the work. Please feel free to let us know if any part of the update could be made clearer.
> > >
> > > Best regards,
> > > Authors of #7282

---

### Official Review · Reviewer_1JVX · 2025-11-11

**Soundness:** 3
**Presentation:** 3
**Contribution:** 3
**Rating:** 6
**Confidence:** 3

**Summary:**

This paper presents RoboOmni, a multimodal large language model framework that infers robot intentions from speech, sounds, and visual cues instead of explicit commands. It introduces a new setting called cross-modal contextual instructions and builds a large dataset, OmniAction, for training. Experiments show RoboOmni outperforms baselines in success rate, speed, and intention recognition in both simulation and real-world tasks.

**Strengths:**

1. Introduces a novel and practical setting, cross-modal contextual instructions, reflecting more natural human-robot interaction without explicit commands.

2. Proposes RoboOmni, a unified framework that integrates intention recognition, interaction confirmation, and action execution using multimodal large language models.

3. Builds a large, diverse dataset (OmniAction) with rich multimodal signals to support training and evaluation.

4. Demonstrates strong empirical performance, outperforming text- and ASR-based baselines in multiple metrics across simulation and real-world environments.

**Weaknesses:**

1. Generalization Beyond Scripted Contextual Cues: A primary concern is the potential for the model to overfit to the specific structures of the six contextual instruction types synthesized for the OmniAction dataset. Since the dataset was generated by prompting GPT-4o to convert atomic instructions into structured dialogues, the model may be learning to recognize these semi-scripted patterns rather than developing a more general, robust capability for open-world intent inference. The impressive performance might not fully transfer to the messiness of real, unscripted human interactions that do not conform to these six categories.

2. Rigidity of the "Infer-Confirm-Act" Protocol: The paper frames the proactive confirmation loop as a key feature. However, this rigid protocol may not always be optimal. A truly intelligent agent should be able to modulate its interaction strategy based on its confidence in its own inference. In unambiguous cases, directly executing the inferred task would be more efficient and fluid. The current framework appears to lack this dynamic capability, which limits its social adaptability.

3. Simplification of the Embodied Action Space: The work's main novelty is in perception and reasoning, while the "Executor" component relies on a standard 7-DoF action representation. The evaluation of success is task-level, which may obscure nuances in the quality of physical execution (e.g., smoothness, safety, precision).

**Questions:**

1. Could you clarify if the Talker and Executor modules are designed to operate sequentially or if they can be interleaved? Can the model generate a sequence that combines speech and action, for instance, to provide narrative feedback during execution (e.g., "Okay, I am now picking up the red cup... a bit heavy... and placing it on the table.")? If not, how do you see this capability being integrated in future work?

2. The use of GPT-4o for dialogue generation is clever and scalable. However, could you discuss potential linguistic artifacts or biases that this synthetic approach might introduce? For instance, did you observe any repetitive conversational structures or phrasings from the LLM that might not be representative of authentic human speech, and how might such artifacts affect the model's real-world performance?

---

> ### Author Response · Authors · 2025-11-25
>
> We sincerely thank you for the thoughtful and constructive review. We appreciate your recognition of our new contextual-instruction setting, unified multimodal framework, large-scale OmniAction dataset, and strong empirical results, and we address the remaining concerns point by point below.
>
> ### W1
> > Generalization Beyond Scripted Contextual Cues: A primary concern is the potential for the model to overfit to the specific structures of the six contextual instruction types synthesized for the OmniAction dataset. Since the dataset was generated by prompting GPT-4o to convert atomic instructions into structured dialogues, the model may be learning to recognize these semi-scripted patterns rather than developing a more general, robust capability for open-world intent inference. The impressive performance might not fully transfer to the messiness of real, unscripted human interactions that do not conform to these six categories.
>
>
> - **The six types were designed as open-ended conversational scenarios, not fixed dialogue patterns.** While several categories (e.g., sentiment, identity, overlapping speech) include task-oriented constraints to probe specific model abilities, the **dyadic and triadic** types only constrain the number of participants, leaving the content and structure fully free. These multi-speaker dialogues therefore emerge naturally and cover a wide spectrum of real human conversational behaviors, mitigating concerns about reliance on scripted patterns.
> - Table 2 shows that RoboOmni performs competitively on **real human–spoken direct instructions**, which are not part of the six contextual types and do not follow any scripted dialogue structure. This demonstrates that the model does not rely on pattern-matching the six categories and can generalize to others instruction styles.
> - We also added a quantitative real-world experiments (Fig 5 in the revision) using **dialogues and speech fully created and recorded by 10 volunteers**, without any templates or LLM involvement. These recordings contain natural disfluencies, hesitation, accents, and unstructured phrasing. RoboOmni’s strong performance in these tests, reaching a 73.9% task success rate and clearly exceeding the best ASR+VLA pipeline (52.2%), demonstrates robust transfer to genuinely unscripted interactions and provides direct evidence against overfitting to synthetic dialogue structures.
>
> | Model        | Sentiment | Non-verbal | Identity | Overlapping | Dyadic | Triadic | Avg  |
> |--------------|-----------|------------|----------|-------------|--------|---------|------|
> | ASR+OpenVLA  | 23.3      | 30.0       | 23.3     | 40.0        | 50.0   | 36.7    | 33.9 |
> | ASR+NORA     | 16.7      | 26.7       | 33.3     | 43.3        | 60.0   | 43.3    | 40.6 |
> | ASR+Pi0      | 46.7      | 36.7       | 40.0     | 53.3        | 70.0   | 66.7    | 52.2 |
> | Ours         | **76.7**      | **66.7**       | **73.3**     | **76.7**        | **80.0**   | **70.0**    | **73.9** |
>
> ### W2
> > 1. Rigidity of the "Infer-Confirm-Act" Protocol: The paper frames the proactive confirmation loop as a key feature. However, this rigid protocol may not always be optimal. A truly intelligent agent should be able to modulate its interaction strategy based on its confidence in its own inference. In unambiguous cases, directly executing the inferred task would be more efficient and fluid. The current framework appears to lack this dynamic capability, which limits its social adaptability.
> - Our benchmark focuses on **contextual instructions**, where prosody, background audio, or multi-speaker dialogue often produces ambiguous intent. In these settings, the “infer–confirm–act” pattern is a necessary strategy to avoid unsafe or incorrect actions, and is a deliberate design choice for the dataset’s goal of proactive assistance.
> - The confirmation step is **not hard-coded**. In scenarios where the user intent is explicit—such as **real human direct instructions** in Table 2—the model executes actions immediately without issuing clarification queries. This demonstrates that RoboOmni naturally adjusts its interaction strategy based on the clarity of the input.

---

> ### Author Response · Authors · 2025-11-25
>
> ### W3
> > Simplification of the Embodied Action Space: The work's main novelty is in perception and reasoning, while the "Executor" component relies on a standard 7-DoF action representation. The evaluation of success is task-level, which may obscure nuances in the quality of physical execution (e.g., smoothness, safety, precision).
>
> - Our action parameterization (EEF pose deltas + gripper) follows the standard practice used in prior VLA systems, including the baselines we compare against, such as OpenVLA[1], OFT[2], NORA[3], π0[4]. This design is widely validated in the community and ensures our model remains compatible with existing datasets, action tokenizers (e.g., FAST+), and evaluation protocols.
> - Existing VLA benchmarks, including LIBERO, π0-series evaluations, primarily use **task-level success** rather than low-level physical motion metrics. Our focus is on **correct intent inference and correct action execution**, which are the core challenges introduced by contextual instructions. Fine-grained metrics such as trajectory smoothness, safety margins, or micro-positioning precision are valuable future directions, but they are orthogonal to the contribution of this paper.
>
> [1] OpenVLA: An Open-Source Vision-Language-Action Model, CORL 2025.
> [2] Fine-Tuning Vision-Language-Action Models, RSS 2025
> [3] Nora: A small open-sourced generalist vision language action model for embodied tasks. Arxiv 2025
> [4] π0 : A vision-language-action flow model for general robot control. Arxiv 2024
>
> ### Q1
> > Could you clarify if the Talker and Executor modules are designed to operate sequentially or if they can be interleaved? Can the model generate a sequence that combines speech and action, for instance, to provide narrative feedback during execution (e.g., "Okay, I am now picking up the red cup... a bit heavy... and placing it on the table.")? If not, how do you see this capability being integrated in future work?
>
> We thank the reviewer for highlighting this valuable future direction.
> - Our current experiments focus on intent inference and task execution rather than continuous narration. Since the OmniAction dataset does not contain interleaved speech–action demonstrations, we do not evaluate this setting in the paper.
> - Although not explored here, RoboOmni’s architecture generates **both text tokens and action tokens in the same token stream**. This means that interleaving speech and action does not require modifying the model architecture—only the data augmentation. In principle, the model could emit sequences like: "`Okay, picking up the red cup. <action_token_x> ... <action_token_x>  Placing it on the table now. <action_token_x> ... <action_token_x>`" because the underlying decoding mechanism is already unified.
> - Enabling real-time narrative feedback therefore mainly depends on providing **training data with interleaved speech–action traces**. The unified AR design makes this a straightforward extension, no new modules or interfaces are needed.
>
>
> ### Q2
> > However, could you discuss potential linguistic artifacts or biases that this synthetic approach might introduce? For instance, did you observe any repetitive conversational structures or phrasings from the LLM that might not be representative of authentic human speech, and how might such artifacts affect the model's real-world performance?
>
> - **We also observed mild repetition in raw GPT-generated dialogues, but we applied multiple steps to reduce artifacts.** To mitigate template-like phrasing or stylistic biases, we applied: (1) **Prompt diversification** to generate varied conversational structures. (2) **Higher-generation temperature** to encourage lexical and syntactic diversity. (3) **ROUGE-L–based redundancy filtering**, removing dialogues with high similarity to existing generated samples. (4) **Human verification** on a subset of dialogues to ensure that resulting speech patterns remain natural and varied. These steps significantly reduce repetitive artifacts and create a distribution closer to real conversational variety.
> - Additionally, the large-scale synthetic data is primarily used to establish robust cross-modal alignment across speech, text, and vision—rather than to mimic human linguistic style. **In practice, GPT-synthetic data does not hinder real-world performance.** Both our **direct human instruction tests** (Tab. 2) and our added **real-speech contextual-instruction experiments** (Fig. 5 in the revision) use fully human-recorded, unscripted speech. RoboOmni maintains strong performance in these settings, indicating that the model does not overfit to GPT-specific phrasing or dialogue structure.

---

> > ### Author Response · Authors · 2025-11-27
> >
> > Dear Reviewer 1JVX,
> >
> > We sincerely appreciate your detailed feedback and the time you've invested in reviewing our paper.  We would like to briefly note that the revision and rebuttal now includes unscripted real-speech contextual-instruction results experiments, clarifies how RoboOmni adapts its confirmation behavior when intent is explicit, and expands the discussion on potential GPT-4o artifacts and our mitigation steps. If anything would benefit from further clarification, we are happy to elaborate.
> >
> > Best regards,
> > Authors of #7282

---

### Author Response · Authors · 2025-11-26
**Summary of Revisions to the Manuscript**

We sincerely thank all reviewers for their time and constructive feedback. Following their valuable comments and suggestions, we have thoroughly revised the paper. All changes are highlighted in blue. The major updates in the newly uploaded version are as follows:
- Section 5.4: We added a real-world contextual-instruction evaluation, where ten volunteers designed contextual dialogues (not GPT-generated) and recorded them across 18 tasks. Our model demonstrates a clear advantage under real-environment, real-speech conditions (73.9% vs. the best baseline at 52.2%) (Figure 5).
- Section 5.6 – Ablation Study: We added modality-drop ablations on vision, audio, and paralinguistic cues for intent recognition. The results show that each component of RoboOmni’s input is essential (Table 3).
- Section 5.6 – Failure Analysis: We included a detailed categorization of error types observed in real-robot experiments (Figure 11).
- Related Work: Added relevant prior works suggested by the reviewers.
- Section 4.1: Added a clear description of the task definition.

We sincerely appreciate the reviewers’ thoughtful input and look forward to further feedback and discussion.

---

### Meta-Review · Area_Chair_wKoz · 2026-01-03

**Summary:**

Across reviewers, RoboOmni is broadly seen as a timely and well-executed paper that introduces an important new setting, cross-modal contextual instructions, and backs it with a large multimodal dataset (OmniAction) and strong empirical results.

**Consistent strengths identified:**
- **Novel problem formulation**: Moving beyond explicit commands toward proactive intent inference from speech prosody, environmental sounds, and vision was widely praised.
- **Unified end-to-end architecture**: Reviewers appreciated the single-stream multimodal token formulation that avoids ASR→text→VLA cascades.
- **Dataset contribution**: OmniAction was viewed as large, diverse, and likely impactful for future research.
- **Empirical performance**: Substantial gains over text- and ASR-based baselines in both simulation and real-robot settings.
- **Clarity and organization**: Most reviewers found the paper well written and easy to follow.

**Recurring concerns across reviewers:**
- Over-reliance on **synthetic data** and limited validation on *real, unscripted contextual speech*.
- Lack of **modality ablations**, making it unclear which cues (audio, paralinguistics, vision) actually drive gains.
- Questions about **novelty vs. integration**, with some reviewers viewing the work as engineering-heavy.
- Limited **evaluation granularity**, especially around safety, dexterity, and over-intervention in proactive behavior.
- Transparency and fairness of **dataset filtering and baselines**.

Overall scores ranged from *marginal accept* to *clear accept*, with disagreement largely hinging on whether the rebuttal evidence sufficiently validated the new setting under real human interaction.

**Reviewer Concerns:**

## Addressed reviewer concerns

**1. Real-speech contextual-instruction validation (multiple reviewers, esp. 1JVX, 2Ge8, xDAu):**
The new setting was validated mainly on TTS/GPT-generated data.

**Rebuttal status:**
Addressed. The authors added:
- A **new real-speech contextual-instruction experiment** with 10 human volunteers, unscripted dialogue, and real robot execution.
- Clear quantitative gains (73.9% vs. 52.2% best baseline).


**2. Modality contribution and ablation studies  (GVRu, 2Ge8):**
Unclear which modalities matter and whether sparse cues (e.g., environmental sounds) are actually used.

**Rebuttal status:**
Addressed through:
- Leave-one-modality-out ablations (no vision / no audio / no paralinguistics).
- Targeted non-verbal sound ablations showing drastic performance drops.
These results clearly demonstrate that gains are not driven by a single modality.


**3. Rigidity of the “infer-confirm-act” loop (1JVX):**
The framework appeared overly rigid and socially inflexible.

**Rebuttal status:**
Addressed. The authors clarified that:
- Confirmation is *not hard-coded*.
- The model executes immediately when intent is clear and only asks clarification under ambiguity.
This resolves a misunderstanding rather than exposing a real limitation.


**4. Dataset filtering, realism, and bias transparency   (xDAu, 1JVX):**
Unclear filtering criteria and risk of synthetic bias.

**Rebuttal status:**
Largely addressed via:
- Detailed explanation of GPT-based visual informativeness filtering.
- Human verification statistics.
- Evidence that real-speech performance does not degrade.


**5. Writing clarity and baseline setup (GVRu, xDAu):**
Insufficient method detail and baseline fairness.

**Rebuttal status:**
Addressed through expanded method sections, clearer task definitions, and explicit clarification of baseline tuning protocols.

---

## Reviewer concerns that remain partially or fully outstanding


**1. Limited evaluation of *proactive safety and over-intervention*  (2Ge8):**
No user study or analysis of false positives, over-eagerness, or misattribution of intent.

**Status:**
Still largely outstanding. The added failure analysis focuses on manipulation errors rather than **social or safety risks** inherent to proactive behavior. This is acknowledged as future work but not empirically addressed.



**2. Depth of algorithmic novelty   (2Ge8):**
The work may lean more toward system integration than new learning principles for proactive intent or uncertainty handling.

**Status:**
The authors clearly articulate *conceptual novelty* (new task + unified token space), but they do not introduce new theoretical mechanisms for uncertainty estimation or decision confidence.


**3. Fine-grained physical execution and safety metrics   (1JVX, 2Ge8):**
Task success may obscure motion quality, safety margins, or dexterity.

**Status:**
Still outstanding. The authors justify their choice by appealing to VLA benchmarking norms, which is reasonable but the concern itself is not resolved, only scoped out.


**4. Compute and environmental cost comparison   (2Ge8):**
Lack of training-time or energy-cost comparison to ASR→VLA pipelines.

**Status:**
Only partially addressed. Inference latency is reported, but full training/energy accounting is not provided.

**Reviewer Scores:**

All reviewers may keep or increase to 6 or above.

---

### Decision · Program_Chairs · 2026-01-26

Accept (Poster)